# Land Surface Model influence on the simulated climatologies of temperature and precipitation extremes in the WRF v.3.9 model over North America

Almudena García-García[1, 2], Francisco José Cuesta-Valero[1, 2], Hugo Beltrami[1], Fidel González-Rouco[3], Elena García-Bustamante[4], and Joel Finnis[5]

[1]Climate & Atmospheric Sciences Institute, St. Francis Xavier University, Antigonish, Nova Scotia, Canada.
[2] Environmental Sciences Program, Memorial University of Newfoundland, St. John's, Newfoundland, Canada.
[3]Physics of the Earth and Astrophysics Department, IGEO (UCM-CSIC), Universidad Complutense de Madrid, Spain.
[4]Research Center for Energy, Environment and Technology (CIEMAT), Madrid, Spain.
[5]Department of Geography, Memorial University of Newfoundland, St. John's, Newfoundland, Canada.

**Correspondence:** Hugo Beltrami (hugo@stfx.ca)

**Abstract.** The representation and projection of extreme temperature and precipitation events in regional and global climate models are of major importance for the study of climate change impacts. However, state-of-the-art global and regional climate model simulations yield a broad inter-model range of intensity, duration and frequency of these extremes. Here, we present a modeling experiment using the Weather Research and Forecasting (WRF) model to determine the influence of the land surface model (LSM) component on uncertainties associated with extreme events. First, we [..[1] ]analyze land-atmosphere interactions within four simulations performed by the WRF model [..[2] ]from 1980 to 2012 over North America, using three different LSMs. Results show LSM-dependent differences at regional scales in the frequency of occurrence of events when surface conditions are altered by atmospheric forcing or land processes. The inter-model range of extreme statistics across the WRF simulations is large, particularly for indices related to the intensity and duration of temperature and precipitation extremes. Our results show that the WRF simulation of the climatology of heat extremes can be $5^oC$ warmer and 6 days longer depending on the employed LSM component, and similarly for cold extremes and heavy precipitation events. Areas showing large uncertainty in WRF simulated extreme events are also identified in a model ensemble from three different Regional Climate Model (RCM) simulations participating in the Coordinated Regional Climate Downscaling Experiment (CORDEX) project, revealing the implications of these results for other model ensembles. Thus, studies based on multi-model ensembles and reanalyses should include a variety of LSM configurations to account for the uncertainty arising from this model component or to test the performance of the selected LSM component before running the whole simulation. This study illustrates the importance of the LSM choice in climate simulations, supporting the development of new modeling studies using different LSM components to understand inter-model differences in simulating temperature and precipitation extreme events, which in turn will help to reduce uncertainties in climate model projections.

---

[1]removed: evaluate

[2]removed: using three different LSMs

## 1 Introduction

General Circulation Models (GCMs) and Regional Climate Models (RCMs) are currently the most useful tools for the study of processes affecting the frequency, duration and intensity of extreme temperature and precipitation events, as well as project-ing their evolution under different emission scenarios at global, regional and local scales. Both observational data and climate
model simulations confirm all of these statistics respond to climate change (Seneviratne et al., 2012; Orlowsky and Seneviratne, 2012; Jeong et al., 2016). However, state-of-the-art global and regional climate models differ substantially in their [..[3] ]repre-sentation of the climatology and response to warming of various indices of temperature and precipitation extremes (Sillmann et al., 2013a, b). Climate information provided by models is currently employed by public and private institutions dedicated to the evaluation and management of risks from extreme events and associated disasters (IPCC, 2013; Arneth, 2019). It is,
therefore, essential that climate models represent extreme events and their evolution as realistically as possible to aid in the design of appropriate policies to mitigate climate change and build resilience. In this study, we [..[4] ]analyze the representation of a set of extreme indices, previously included in international reports such as the IPCC (2013) and Seneviratne et al. (2012), as simulated by the Weather Research and Forecasting (WRF) model with different land surface model (LSM) components. We focused on the climatology of these extreme indices, that is the mean of of each index from 1980 to 2013.
[..[5] ]Soil conditions are coupled to near-surface atmospheric phenomena through energy and water exchanges at the ground surface. The representation of the interactions between the land surface and the near-surface atmosphere has been identified as a key factor in the simulation of extreme events (e.g. Lorenz et al., 2016; Vogel et al., 2017). [..[6] ]For example, [..[7] ]changes in soil moisture and soil properties may lead to variations in energy fluxes at the land surface affecting temperature and precipitation evolution. Changes in latent heat flux affect surface temperatures in the following way: a
decrease in latent heat flux likely means an increase in the energy available for sensible heat flux, which is directly related to the air-ground temperature gradient. The increase in sensible heat flux yields an increase in this temperature gradient, which may lead to changes in air temperatures (Seneviratne et al., 2010). Meanwhile, changes in latent heat flux also yield changes in the atmospheric water content, possibly affecting the formation of clouds and precipitation (Seneviratne et al., 2010). Previous observational studies have shown the impact of soil moisture deficits on hot extreme temperatures
through changes in evapotranspiration over southeastern and western Europe and Russia (Hirschi et al., 2011; Miralles et al., 2012; Hauser et al., 2016). Additionally, soil moisture regimes have been found to alter the energy and water exchanges at the surface, influencing inter-annual summer temperature variability in central parts of North America (Donat et al., 2016), and precipitation events in western North America (Diro et al., 2014). Land-Atmosphere interactions, and consequently near-

---

[3] removed: interpretation

[4] removed: evaluated

[5] removed: Land-atmosphere interactions have

[6] removed: Soil conditions affect and are affected by near-surface atmospheric phenomena, through energy and water exchanges at the ground surface .

[7] removed: previous

surface conditions, are influenced by vegetation and snow covers (Stieglitz and Smerdon, 2007; Diro et al., 2018). For example, Diro et al. (2018) showed that interactions between snow cover and atmospheric processes influence extreme events, increasing the frequency of cold events over western North America and affecting the variability in warm events over northeast Canada and the Rocky mountains.

Metrics built on the representation of land-atmosphere interactions have been employed as a basis for evaluating extreme temperature and precipitation events in climate model simulations (Knist et al., 2016; Davin et al., 2016; Lorenz et al., 2016; Sippel et al., 2017; Gevaert et al., 2018; García-García et al., 2019). For example, Lorenz et al. (2016) evaluated outputs from six GCMs participating in the Global Land-Atmosphere Coupling Experiment of the Coupled Model Intercomparison Project, Phase 5 (GLACE-CMIP5) and concluded that ranges of intensity, frequency and duration of extreme events among climate projections are strongly related to inter-model differences in the representation land-atmosphere interactions. Gevaert et al. (2018) evaluated the representation of land-atmosphere interactions within a set of [..[8] ]offline LSM simulations, finding similar spatial patterns of soil moisture-temperature coupling among LSM simulations, but large variability in the degree and local patterns of land-atmosphere coupling. García-García et al. (2019) employed a simple metric derived from soil and air temperatures to evaluate outputs from the CMIP5 models [..[9] ]against observations over North America, suggesting a strong dependency of the simulated land-atmosphere interactions on the LSM component employed. The model differences in the representation of land-atmosphere interactions shown in these studies may be affecting the simulation of extreme events, and thus contributing to the uncertainty in multi-model ensembles such as those formed by the CMIP5 and the Coordinated Regional Climate Downscaling Experiment (CORDEX) simulations.

[..[10] ]The choice and complexity of the LSM component may have implications for the representation of land-atmosphere interactions in reanalysis products, since reanalysis products have shown discrepancies in the representation of land-atmosphere coupling with observations (Ferguson et al., 2012; García-García et al., 2019). However, in contrast with the variety of LSM components employed in the new generation of GCMs, reanalyses use simplified versions of LSM components, typically included as part of the atmospheric model component. For example, all reanalysis products produced by the European Centre for Medium-range Weather Forecasts (ECMWF) model (CERA-20C, ERA-15, ERA20C, ERA-Interim and ERA-40 products) employed the same LSM component included in the code of the ECMWF atmospheric model. The two Modern-Era Retrospective analysis for Research and Applications (MERRA) global products employed the GEOS-5 Catchment land surface model (Reichle et al., 2011). The Japanese Reanalysis (JRA) products employed a modified version of the Simple Biosphere (SiB) LSM (Onogi et al., 2007), while most of National Centers for Environmental Prediction (NCEP) and National Center for Atmospheric Research (NCAR) products employed the NOAH LSM (Tewari et al., 2004). The complexity and variety of these LSM components are limited in order to reduce computational costs, affecting the quality of the represented land surface processes. This has already been noted by the scientific community, and some have attempted to address the issue by incorporating updated versions of LSMs in new land reanalysis products though offline LSM simulations forced

---

[8]removed: off-line

[9]removed: and the North American Regional Reanalysis (NARR)

[10]removed: In

by observational data products (LDAS, MERRA-land, ERA-Iterim/Land, Rodell et al., 2004; Reichle et al., 2011; Balsamo et al., 2015). Although these new products can be useful for LSM development and provide data about the soil states and fluxes (Balsamo et al., 2015), the offline character of the new land products inhibits the representation of land-atmosphere coupling and feedbacks.

Here, we perform a set of modeling experiments to [..[11] ]examinate for the first time the influence of the LSM component on the simulation of key extreme indices and land-atmosphere interactions within land-atmosphere coupled climate simulations at continental scales. For this purpose, four regional simulations are performed over North America (1979-2012) using the WRF model including three different LSM components widely employed in model simulations and reanalysis products, as described in Section 2. To explore the influence of the LSM component on the simulation of extreme events in multi-model ensembles, we compare the uncertainty in the representation of extreme indices within our four WRF simulations with the uncertainty in three simulations participating in the North American component of the CORDEX project (NA-CORDEX). The methodology for the analysis of land-atmosphere interactions and the representation of extreme events is described in Section 3. Section 4 presents the [..[12] ]examination of land-atmosphere interactions, the analysis of LSM differences in the representation of temperature and precipitation extremes, and the comparison between the WRF simulations and three [..[13] ]CORDEX simulations. A discussion about previous results and the main conclusions and implications of this study are presented in Section 5 and 6, respectively.

## 2  Description of the modeling experiment

We performed four regional simulations over North America (NA) using the version 3.9 of the Advanced Research WRF (ARW-WRF) model (Michalakes et al., 2001) including three different land surface models: the NOAH LSM (NOAH, Tewari et al., 2004), the NOAH LSM with multiparameterizations options (NOAH-MP, Niu et al., 2011), and the Community Land Model version 4 LSM (CLM4, Oleson et al., 2010). Vegetation cover was prescribed in these three simulations (NOAH, NOAH-MP and CLM4); an additional simulation was conducted with dynamic vegetation cover in the NOAH-MP LSM (NOAH-MP-DV), allowing for the evaluation of the influence of dynamic vegetation on extremes[..[14] ]. The use of different LSM [..[15] ]configurations in a RCM permits the study of the influence of surface and soil processes on the simulated climate system in contrast to LSM offline simulations (Laguë et al., 2019).

The LSM components employed have been previously included in climate model studies or in reanalysis products. The CLM4 LSM component has been coupled to several GCMs participating in the CMIP5 project (Collins et al., 2006; Vertenstein et al., 2012). The NOAH LSM has been extensively used for reanalysis products, as well as for RCM simulations such as those participating in the CORDEX project (Mesinger et al., 2006; Katragkou et al., 2015). The NOAH-MP LSM has been selected

---

[11]removed: evaluate

[12]removed: evaluation

[13]removed: Coordinated Regional Climate Downscaling Experiment (CORDEX ) Evaluation

[14]removed: (NOAH-MP-DV)

[15]removed: components

**Table 1.** Characteristics of the LSM components employed for the WRF simulations performed in this analysis.

| LSM | Vegetation Types | Vegetation Mode | Soil Layers | Soil Depth | Snow | Reference |
|---|---|---|---|---|---|---|
| NOAH | Dominant vegetation type in one grid cell | Prescribed | 4 | 2 m | Single Layer | Tewari et al. (2004) |
| NOAH-MP | Dominant vegetation type in one grid cell | Prescribed | 4 | 2 m | Up to 3 Layers | Niu et al. (2011) |
| NOAH-MP-DV | Dominant vegetation type in one grid cell | Dynamic | 4 | 2 m | Up to 3 Layers | Niu et al. (2011) |
| CLM4 | Up to 10 vegetation types in one grid cell | Prescribed | 10 | 4.32 m | Up to 5 Layers | Oleson et al. (2010) |

for current studies using WRF (e.g. Liu et al., 2017). The NOAH LSM is a rather basic LSM developed by the National Center for Atmospheric Research (NCAR) and the National Centers for Environmental Prediction (NCEP), based on the Oregon State University (OSU) LSM (Mitchell, 2005). This LSM component describes soils using 4 layers with thickness 10, 30, 60 and 100 cm, using a zero-flux bottom boundary condition at a depth of 2 m. The NOAH LSM estimates soil moisture and temperature at the node of each soil layer, taking into account snow cover, canopy moisture, and soil ice. The NOAH-MP

LSM is based on the NOAH LSM, introducing relevant improvements, such as a dynamic vegetation option; a new separated vegetation canopy cover that improves the computation of energy, water and carbon fluxes at the surface; a separate scheme for computing energy fluxes over vegetated surfaces and bare soils; a new 3-layer snow model; a more permeable frozen soil; and an improved description of runoff and soil moisture. Although the NOAH-MP LSM is the updated version of the NOAH LSM and has been shown to improve the simulation of surface processes in comparison to the NOAH LSM (e.g. Niu et al., 2011;

Yang et al., 2011), the NOAH-MP LSM has not yet been implemented in any reanalysis product. The CLM4 represents one of the most advanced LSM components, incorporating a detailed description of biogeophysics, hydrology and biogeochemistry. The CLM4 classifies vegetation cover [..[16] ]using up to 16 different plant functional types, considering the physiology and structure of different plants. The soil vertical structure is divided into a layer for the vegetation canopy, 5 layers for snow cover, and 10 soil layers, placing the zero-flux bottom boundary condition at approximately 4.32 m. The main characteristics of the

employed LSM components are summarized in Table 1.

Beyond the structural differences among LSM components, the remaining options and parameters are identical for the four WRF simulations. Boundary conditions for the WRF experiments are provided by the North American Regional Reanalysis (NARR) product, which is formed by the NCEP Eta atmospheric model, the NOAH LSM and the Regional Data Assimilation System (RDAS); (Mesinger et al., 2006). NARR data are provided with a 32 km grid and three-hourly temporal resolution,

available at the National Center for Environmental Information (NOAA) archive. The domain set for the WRF simulations has 50 km horizontal resolution and 27 atmospheric levels, covering North America in a Lambert projection. The land use categories employed for the four simulations (Figure S1 in the supplementary information) are derived from the Moderate Resolution Imaging Spectroradiometer (MODIS, Barlage et al., 2005). Sea surface temperatures were prescribed using the NARR product. The four WRF simulations start [..[17] ]on January 1st 1979, which is the first year of the NARR product,

---

[16]removed: according to 4

[17]removed: in

and end [..[18] ]on December 31st 2012, using a time-step of 300 seconds for the model integrations. We use the first year of each simulation as spin-up and the other 33 years for the analysis. The selection of the first year as spin-up was done considering the initialization period previously used in WRF climate experiments, such as those in Wang and Kotamarthi (2015); Katragkou et al. (2015); Barlage et al. (2005). The comparison of the latent heat flux and surface air temperature outputs from the WRF-CLM4 simulation starting on January 1st, 1979 and a similar simulation starting on June 1st, 1979

indicates that this period is enough to initialize the simulation (Figures S22 and S23 in the supplementary information). The employed physics parameterizations include the WSM 6-class graupel scheme for the microphysics (Hong and Lim, 2006), the Grell-Freitas ensemble scheme for cumulus description (Grell and Freitas, 2014), the Yonsei University scheme as planetary boundary layer scheme (YSU, Hong et al., 2006), the revised MM5 monin-Obukhov scheme for the surface layer (Jiménez et al., 2012), and the CAM scheme for the integration of radiation physics each 20 min intervals (Collins et al., 2004).

The gap in resolution from the employed boundary conditions (32 km) to the final simulations (50 km) can be counter-intuitive for a RCM experiment[..[19] ]. The computational resources saved with this coarse resolution allow us to perform simulations long enough for the [..[20] ]study of land-atmosphere interactions and extreme events at climatological scales and yet similar horizontal resolution and domain to those employed in the North American component of the CORDEX project (Giorgi and Gutowski Jr., 2015) can be attained. Thus, this decrease in resolution allows us to generate a set of four WRF

sensitivity experiments using different LSM configurations. Additionally, we do not apply any nudging technique, ensuring that the RCM evolves freely according to each LSM component and its representation of land-atmosphere interactions.

## 3   Methodology

Different metrics have been employed in the literature for the evaluation of land-atmosphere interactions within climate model simulations and observations. Among these metrics, we selected the Vegetation-Atmosphere Coupling (VAC) index (Zscheis-

chler et al., 2015) as our evaluation metric for the representation of land-atmosphere interactions at monthly scales. This index has been previously employed in the literature to identify regions with episodes of strong land-atmosphere coupling within climate model simulations and observational data (Zscheischler et al., 2015; Gevaert et al., 2018; Sippel et al., 2017; Li et al.,

---

[18]removed: in

[19]removed: ; indeed. The rationale for this decrease in resolution is that this set of simulations constitutes an ensemble of WRF sensitivity experiments to using different LSM components. The

[20]removed: evaluation

2017; Philip et al., 2018). The VAC index is segregated in four categories based on the simultaneous occurrence of some given extreme percentile rages of Surface Air Temperature (SAT) and latent heat flux (LH, Philip et al., 2018):

$$
\begin{array}{llllll}
VAC_a & if & SAT < 30^{th}Pctl. & and & LH < 30^{th}Pctl. \rightarrow Atmo. & Control & Coupling \\
VAC_b & if & SAT > 70^{th}Pctl. & and & LH > 70^{th}Pctl. \rightarrow Atmo. & Control & Coupling \\
VAC_c & if & SAT > 70^{th}Pctl. & and & LH < 30^{th}Pctl. \rightarrow Land & Control & Coupling \\
VAC_d & if & SAT < 30^{th}Pctl. & and & LH > 70^{th}Pctl. \rightarrow Land & Control & Coupling \\
0 & otherwise
\end{array}
\tag{1}
$$

Extremes of SAT and LH are defined as values exceeding (below) the 70th (30th) percentile, relative to a 20-year period (1980-2000) (Eq. 1). [..[21] ]We use the VAC metric at monthly scales as in Sippel et al. (2017), since this work proved the usefulness of the VAC metric at monthly time scales for the analysis of the climatology of extreme indices. The VAC index classifies areas depending on the soil moisture regime into energy-limited areas, where atmospheric [..[22] ]conditions controls land-atmosphere interactions (VAC$_a$ and VAC$_b$), and [..[23] ]into water-limited areas, where [..[24] ]soil moisture deficits control the water and energy exchanges at the air-ground interface (VAC$_c$ and VAC$_d$). As explained in Zscheischler et al. (2015), the VAC$_a$ category is associated with [..[25] ]energy limitations (low SAT) caused by the presence of clouds and precipitation, which leads to [..[26] ]a decrease in the vegetation photosynthetic activity and therefore an increase in soil moisture. The VAC$_b$ category is frequent in wet areas with high SAT, usually related to clear sky and high radiation, which is associated with [..[27] ]an increase in the vegetation photosynthetic activity inducing the depletion of soil moisture. During VAC$_c$ episodes, the combination of high SAT and soil moisture deficits leads to diminished vegetation photosynthetic activity, followed by low precipitation and consequently [..[28] ]low soil moisture and high SAT, promoting heat waves and droughts. The VAC$_d$ category is associated with high precipitation over dry soils which stimulates vegetation photosynthetic activity, increases soil moisture and decreases SAT. A no-coupling option also occurs when SAT and LH extremes do not coincide in time.

We calculate the frequency of occurrence for each VAC category using deseasonalized and detrended monthly SAT and LH time series following the typical methodology (Sippel et al., 2017) at each grid cell from 1980 to 2012, hereafter the analysis period. The frequency of occurrence for each VAC category is calculated by counting the VAC events for the analysis period seasonally; in boreal winter (December, January, and February; DJF), in spring (March, April, and May; MAM), in summer (June, July, and August; JJA), and in fall (September, October, and November; SON). The [..[29] ]probability of each VAC category (Figures S2-S5 in the supplementary information) and the probability of the no-coupling case sum 100% over

---

[21]removed: The 30th and 70th percentile thresholds were also employed in previous studies based on monthly data (Sippel et al., 2017)

[22]removed: forcing controls processes at the land surface

[23]removed: transitional

[24]removed: land surface processes are driven by

[25]removed: low SAT

[26]removed: low vegetation activity likely rising

[27]removed: the increase in vegetation

[28]removed: reduced

[29]removed: VAC frequencies of

the analysis period at each grid cell. The VAC probabilities of occurrence for each category are considered significant when higher than the 95th percentile of the population obtained by 100 randomly sorted 34-year time series of SAT and LH. For the study of land-atmosphere coupling within each simulation, we represent the averaged frequency of events under atmospheric control ($VAC_a$ and $VAC_b$) and under land control ($VAC_c$ and $VAC_d$) at grid cells with significant frequency of occurrence for at least one of the two VAC categories.

After the [..[30] ]analysis of land-atmosphere interactions in our set of simulations, we assess the representation of extreme events across the WRF simulations coupled to different LSM components. There are several definitions of indices related to temperature and precipitation extremes, mainly using thresholds based on absolute values or statistical percentiles (e.g. Sillmann et al., 2013a). [..[31] ]Studies based on statistical percentiles improve the comparison among models but hamper the interpretation of results by losing the physical meaning of the variable (temperature or precipitation). Although the use of extreme indices defined with absolute values facilitates the understanding of results by a general public, these indices could include model-specific biases[..[32] ]. These biases can be corrected by bias removal techniques[..[33] ], however, the advantage of applying bias removal techniques [..[34] ]is not clear for the study of future climate trends and climate variability, since [..[35] ]these techniques have been proven to modify the spatiotemporal consistency of climate models as well as internal feedback mechanisms and conservation terms (Ehret et al., 2012; Cannon et al., 2015). Additionally, the simulation of absolute temperatures are of central importance for temperature dependent processes that may have important consequences for society and ecosystems, such as soil carbon processes (Hicks Pries et al., 2017). [..[36] ]Since extreme indices based on both absolute values and statistical thresholds present advantages and disadvantages, we selected a set of indices including both categories from the list of 27 indices recommended by the Expert Team on Climate Change Detection and Indices (ETCCDI, Karl et al., 1999, Table 2). The employed intensity indices of temperature events are based on temperature values in the hottest day and coldest night in summer and winter for warm and cold events. The frequency indices of the same events indicate the percentage of hot and cold days and nights in the year. The duration of the temperature events is represented with the number of consecutive hot days and cold nights. The intensity of heavy precipitation events is characterized by the total annual precipitation in wet days, while the frequency of precipitation events is studied using the number of very wet days per year. The duration of wet and dry events is represented with the annual number of consecutive wet and dry days. For more specific definitions of the indices employed in this study, please refer to Table 2. Since we are interested in the climatology of extreme events, temporal averages of each annual index are computed for the analysis period at each grid cell for each WRF experiment. Then, we compute the inter-model range of each index across the WRF simulations (i.e., the

---

[30]removed: evaluation

[31]removed: The evaluation of model simulations in representing indices based on absolute values

[32]removed: , that

[33]removed: . However,

[34]removed: techniques

[35]removed: they

[36]removed: Studies based on statistical percentiles improve the comparison among models but hamper the interpretation of results by losing the physical meaning of the variable (temperature or precipitation).

**Table 2.** List of extreme indices used in this study defined by the Expert Team on Climate Change Detection and Indices (ETCCDI) (Karl et al., 1999). Percentiles are calculated over the period 1980-2000.

| Index | Definition | Unit |
|---|---|---|
| **Cold Event** | | |
| **Intensity** | | |
| TXx DJF | Maximum value of daily maximum temperature (hottest day) in winter | $^\circ C$ |
| TNn DJF | Minimum value of daily minimum temperature (coldest nigth) in winter | $^\circ C$ |
| **Frequency** | | |
| TN10p | Percentage of days in a year when daily minimum temperature | |
| | < the calendar day 10th percentile centered on a 5-day window | % |
| TX10p | Percentage of days in a year when daily maximum temperature | |
| | < the calendar day 10th percentile centered on a 5-day window | % |
| **Duration** | | |
| CSDI | Cold Spell Duration Index: annual count of days with at least 6 consecutive days when | |
| | daily minimum temperature < the calendar day 10th percentile centred on a 5-day window | Days |
| **Warm Event** | | |
| **Intensity** | | |
| TXx JJA | Maximum value of daily maximum temperature (hottest day) in summer | $^\circ C$ |
| TNn JJA | Minimum value of daily minimum temperature (coldest night) in summer | $^\circ C$ |
| **Frequency** | | |
| TN90p | Percentage of days in a year when daily minimum temperature | |
| | > the calendar day 90th percentile centered on a 5-day window | % |
| TX90p | Percentage of days in a year when daily maximum temperature | |
| | > the calendar day 90th percentile centered on a 5-day window | % |
| **Duration** | | |
| WSDI | Warm Spell Duration Index: annual count of days with at least 6 consecutive days when | |
| | daily maximum temperature > the calendar day 90th percentile centred on a 5-day window | Days |
| **Precipitation Event** | | |
| **Intensity** | | |
| R95p | Annual total precipitation when daily accumulated precipitation on a wet day | |
| | > 95th percentile of precipitation on wet days | mm |
| **Frequency** | | |
| R10mm | Annual count of days when daily accumulated precipitation $\geq$ 10mm | Days |
| **Duration** | | |
| CDD | Maximum length of dry spell: maximum annual number of consecutive days with daily | |
| | accumulated precipitation < 1mm | Days |
| CWD | Maximum length of wet spell: maximum annual number of consecutive days with daily | |
| | accumulated precipitation $\geq$ 1mm | Days |

difference between the maximum and minimum values at each grid cell considering the four WRF simulations), using it as metric for the uncertainty in the WRF simulation of extreme events arising from the LSM component.

The [..[37] ]effect of the LSM configuration on the simulation of extreme events can also be relevant for multi-model ensembles, such as those participating in the CORDEX project. Here, we compare the LSM effect on the WRF simulation of extreme temperature and precipitation events [..[38] ]with the representation of extreme events by three different RCMs participating in the North America CORDEX (NA CORDEX) program [..[39] ](Mearns et al., 2017). For this purpose, we use the daily outputs from three NA-CORDEX simulations forced by reanalysis data (Evaluation experiments, Table 3). These CORDEX simulations were performed by the WRF model (Skamarock et al., 2008), the RCA4 model (Samuelsson et al., 2011), and the CRCM-UQAM model (Martynov et al., 2013), using boundary conditions from the ERA-Interim reanalysis (Dee et al., 2011). The remaining NA-CORDEX Evaluation simulations available in the Climate Data Gateway at NCAR were not used because those simulations cover a significantly shorter period of time than our simulations. The spatial domain and resolution of the NA CORDEX simulations are similar to that of the WRF simulations, as indicated in Section 2. Refer to Table [..[40] ]S1 for information about the availability of the data employed in this work.

**Table 3.** Characteristics of the Evaluation simulations employed in this analysis from three RCMs participating in the NA-CORDEX project. The boundary conditions for these three simulations are from the ERA-Interim reanalysis.

| CORDEX RCM | LSM | Vegetation Types | Spectral Nudging | Institution | Reference |
|---|---|---|---|---|---|
| WRF | NOAH | 24 | Yes | NCAR | Skamarock et al. (2008) |
| RCA4 | RCA LSS | 12 | No | SMHI | Samuelsson et al. (2011) |
| CRCM-UQAM | CLASS3.5+ | 4 | No | UQAM | Martynov et al. (2013) |

# 4  Results

## 4.1  [..[41] ]Examination **of land-atmosphere interactions in WRF simulations**

All WRF simulations with different LSM components display similar spatial patterns for VAC categories, agreeing in the seasonality and broadly in the [..[44] ]classification into energy and water limited areas, that is, in the areas with high probability of episodes when atmospheric forcing or soil conditions control [..[45] ]land-atmosphere interactions (Figures 1 and 2). Atmospheric forcing controls surface processes at middle and high latitudes in MAM, JJA and SON, moving southward in DJF

---

[37] removed: LSM

[38] removed: was also compared

[39] removed: , using daily data from three Evaluation simulations (Table S1

[40] removed: S2

[41] removed: Evaluation

[44] removed: areas

[45] removed: processes at the land surface

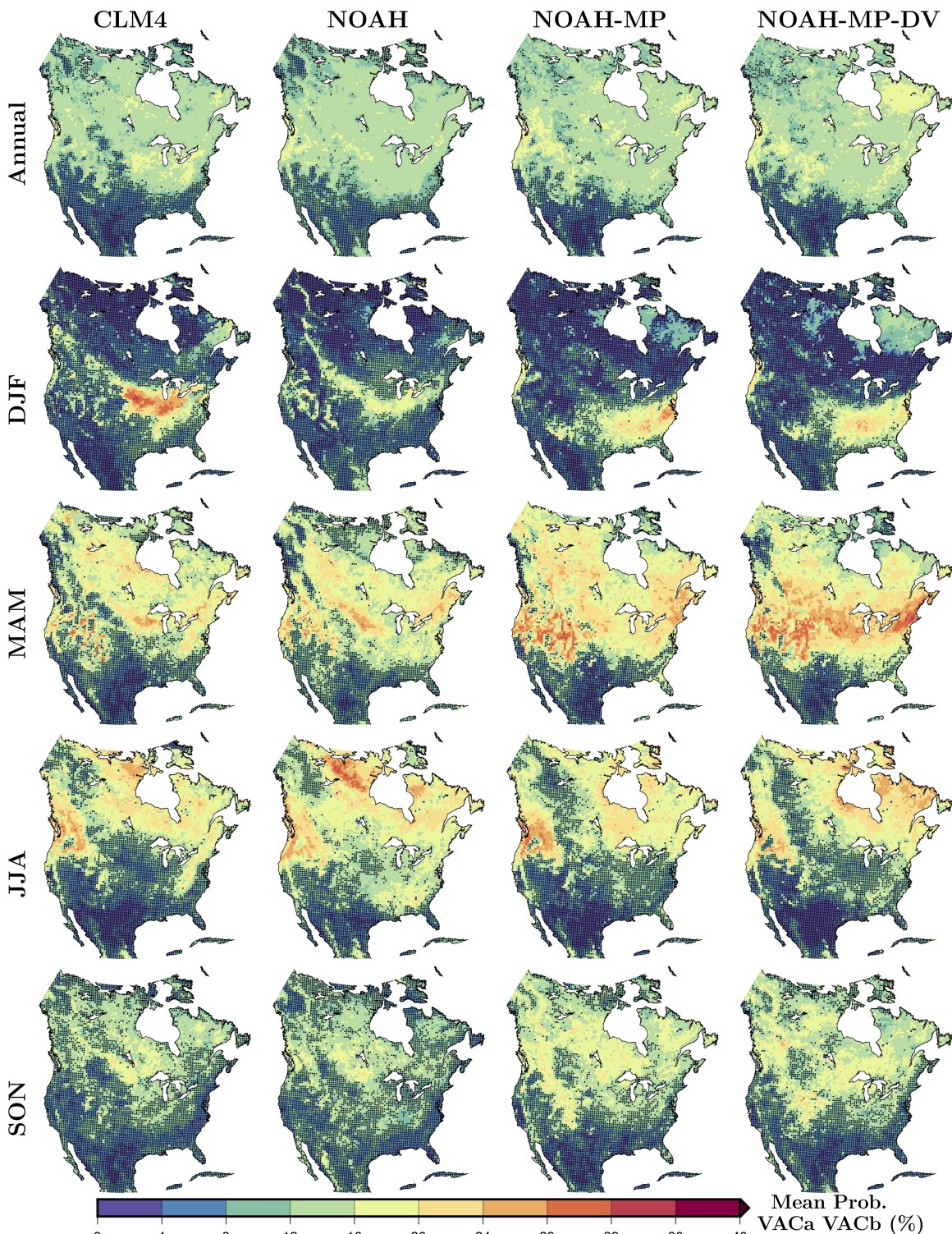

**Figure 1.** Mean frequency of occurrence for VAC categories associated with atmospheric control (VAC$_a$ and VAC$_b$) for each simulation annually and seasonally; DJF, MAM, JJA and SON. Black dots in the maps indicate VAC values [..$^{42}$ ]lower than the 95th percentile of the randomly generated series, and therefore areas with no significant probabilities.

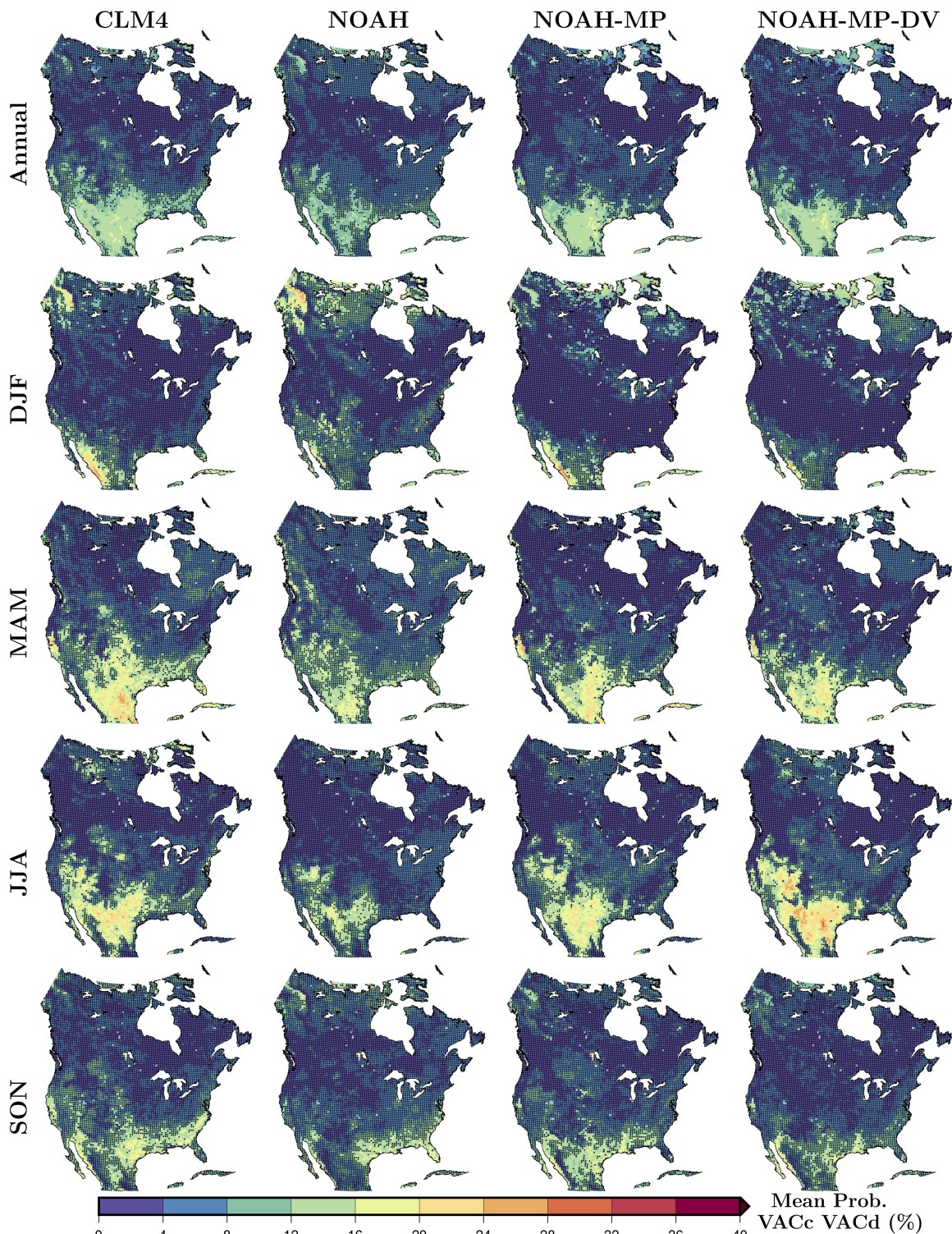

**Figure 2.** Mean frequency of occurrence for VAC categories associated with land control (VAC$_c$ and VAC$_d$) for each simulation annually and seasonally; DJF, MAM, JJA and SON. Black dots in the maps indicate VAC values [..[43] ]lower than the 95th percentile of the randomly generated series, and therefore areas with no significant probabilities.

(Figure 1). Areas frequently driven by soil [..[46] ]conditions are displayed over the western Mexican coast in DJF, spreading across low and middle latitudes in MAM, JJA and SON (Figure 2). These spatial similarities in the VAC coupling metric indicate that factors common in our four simulations, such as land cover, topography, latitudinal differences or atmospheric parameterizations produce these spatial patterns. Despite the broad agreement between LSM simulations in the spatial distribution of the VAC categories, there are regional differences in their representation of land-atmosphere coupling. These regional differences allow us to identify the NOAH LSM as the one simulating the weakest annual land control on processes at the surface, mainly due to a relatively weak land control during MAM and JJA (Figure 2).

The areas where LSM simulations differ in the probability of episodes under atmospheric control ($VAC_a$ and $VAC_b$) vary with the season; for example the NOAH-MP LSM simulates a large area under atmospheric control over the southeastern US in DJF, while the CLM4 and NOAH LSMs identify atmospheric control areas below the Great Lakes following a northwestern direction (Figure 1). These differences in atmospheric control areas are caused by the different probability of extreme latent heat flux simulated by each LSM in DJF (Figure S6 and S7 in the supplementary information). In MAM, the NOAH-MP LSM represents higher probability of atmospheric control episodes over the northern US in comparison with the CLM4 and NOAH simulations (Figure 1). The NOAH simulation shows the strongest atmospheric control in JJA as compared with the remaining simulations, particularly over eastern and western regions of Hudson Bay, the southeastern US and small areas in Mexico (Figure 1). This strong JJA atmospheric control in the NOAH simulation is driven by the $VAC_a$ category (Figure S2 in the supplementary information), and likely related to the high probability of cold temperatures over these areas in this simulation (Figure S9 in the supplementary information). During SON, the NOAH-MP LSM reaches the highest probability of episodes under atmospheric control at middle and high latitudes, caused by the high probability of extreme latent heat flux in comparison with the rest of the LSMs (Figure S6 and S7 in the supplementary information). The contribution of the $VAC_a$ and $VAC_b$ categories to these episodes is broadly similar across LSMs, with slightly higher $VAC_a$ in all seasons; modest LSM-specific differences include a tendency for the NOAH simulation to show slightly higher $VAC_a$ probabilities across all seasons (but especially DJF) (Figures [..[47] ]S2 [..[48] ]and S3).

Although the NOAH simulation displays the weakest land control for all seasons, it shows regions under land control over northwestern North America in DJF also indicated by the CLM4 simulation, but absent in the NOAH-MP and NOAH-MP-DV simulations (Figure 2). The probability of land control episodes over the western Mexican coast is higher in the CLM4 and NOAH-MP simulations than in the NOAH and NOAH-MP-DV simulations in DJF. These LSM differences are associated with the high probability of low latent heat flux over those regions in winter for the CLM4 and the NOAH-MP simulations in comparison with the remaining simulations (Figure S7 in the supplementary information). In JJA, however, the NOAH-MP-DV simulation presents a stronger land control at low and middle latitudes than the NOAH-MP simulation (Figure 2), mainly caused by the $VAC_d$ category and the high probability of cold temperatures (Figures S5 and S9 in the supplementary information). There are also regional differences between LSM simulations in SON, particularly over the southeastern US

---

[46]removed: processes

[47]removed: S1 and

[48]removed: ). LSM differences in the representation of $VAC_a$ and $VAC_b$ probabilities suggest the LSM influence on the evolution of atmospheric conditions.

coast where the CLM4 shows the strongest land control, followed by the NOAH-MP simulation (Figure 2). The NOAH-MP-DV simulation do not show this strong land control at low latitudes in SON, due to the low probability of high latent heat flux represented by the NOAH-MP LSM with dynamic vegetation (Figure S6 in the supplementary information). The weaker land control in the NOAH simulation, however, is not explained by the probability of extreme temperature or latent heat flux, since these probabilities are similar to those in the CLM4 simulation (Figures S6-S9 in the supplementary

information). Thus, it is associated with the absent of coincidences of extreme temperature and latent heat flux simulated by the NOAH LSM. Exploring the contribution of [..[49] ]$VAC_c$ and $VAC_d$ separately, it is shown they present small differences; for example, the $VAC_c$ probability in DJF is slightly higher than the $VAC_d$ probability for all simulations, showing the opposite behavior in JJA for the NOAH-MP and the NOAH-MP-DV simulations (Figures [..[50] ]S4 [..[51] ]and S5 in the supplementary information).

**4.2** Climatologies of temperature and precipitation extremes in the WRF simulations

We continue this analysis comparing the representation of [..[52] ]extreme events within the four WRF simulations by calculating the range among these four simulations. But first, we analyze the spatial features of the climatology of extreme temperature and precipitation indices as simulated by the mean of the four WRF simulations with different LSM configurations (hereafter WRF ensemble mean) and by each LSM simulation separately.

The climatologies of temperature and precipitation [..[53] ]extreme indices as described in Table 2 and represented by the mean of each index for the analysis period, [..[54] ]show similar spatial patterns across all WRF simulations with different LSM [..[55] ]configurations (Figures S10, S11 and S12 in the supplementary information). The similarities in the spatial pattern of extreme events among our simulations indicate that other factors different from the LSM configuration, such as land cover, topography, latitudinal differences and atmospheric parameterizations, are driven these spatial features. Figure 3

represents the simulated climatologies of all extreme indices for the ensemble mean, formed by [..[56] ]the four WRF simulations. The WRF ensemble mean shows the most intense cold events at high latitudes and high elevations, with cold events being more frequent and longer over northwestern North America and over Mexico (Figure 3a). The simulation of warm events is more intense in coastal areas of the US and Mexico and over the central US, being more frequent and longer over southern North America with a high percentage of hot nights over northeastern NA (Figure 3b). Precipitation events are heavier and more

frequent at higher elevations and over southeastern NA (Figure 3c). The longest dry periods are simulated over the western

---

[49]removed: respective

[50]removed: S3 and

[51]removed: ). The LSM differences shown in

[52]removed: land control VAC categories likely imply LSM differences in the simulated statistic of extreme events because of the relationship between $VAC_c$ episodes and heat waves and droughts (Zscheischler et al., 2015)

[53]removed: extremes

[54]removed: represented by their means,

[55]removed: component (Figures S5, S6 and S7).

[56]removed: all

Mexican and US coasts, reaching more than 80 consecutive dry days, while the longest wet periods are represented over the Rockies and the northwestern Mexican coast (Figure 3c).

Figure 4 summarizes the averaged climatology of each extreme index for each simulation. Averages are computed over six regions adapted from Giorgi and Francisco (2000): Central America, CAM; Western North America, WNA; Central North America, CNA; Eastern North America, ENA; Alaska, ALA; and Greenland, GRL. Colors in the figure correspond to the hottest (red) and coldest (blue) index values among the WRF simulations for the representation of cold and warm temperature extremes, and to the driest (brown) and wettest (green) index values for the representation of precipitation extremes over each region. This approach helps us to identify the CLM4 simulation as that with the weakest and shortest cold extreme events, although simulating more frequent cold events than the rest of the LSM components (Figure 4a). Meanwhile, the NOAH-MP-DV simulation shows more intense cold extremes during shorter periods over most of the regions (CAM, CNA, ENA and ALA) in comparison with the NOAH-MP simulation which uses prescribed vegetation (Figure 4a). The CLM4 simulation also corresponds to the most intense representation of warm extremes for the index based on maximum temperatures, while the intensity index based on minimum temperatures shows higher values in the NOAH-MP simulation, except for the CAM region (Figure 4b). The NOAH simulation is associated with the weakest and shortest warm extremes over most areas, and the NOAH-MP and NOAH-MP-DV simulations with the most frequent and longest events. The effect of dynamic vegetation seems to weaken hot extremes at nights over all regions, making them longer at middle and high latitudes (CNA, ENA, ALA and GRL), except in the western US (Figure 4b). That is, the NOAH-MP-DV simulation yields warm events longer but not as hot as using prescribed vegetation at most regions at middle and high latitudes. For precipitation extreme events, the CLM4 simulation shows the most intense and frequent precipitation events over most areas, while the NOAH simulation shows the weakest and the least frequent precipitation events (Figure 4c). The NOAH-MP simulation produces the longest dry periods over all regions except at high latitudes, where the NOAH-MP-DV simulation yields a higher number of consecutive dry days (Figure 4c). The simulation with dynamic vegetation yields wetter results than the simulation with prescribed vegetation at middle and low latitudes, while at high latitudes the NOAH-MP-DV simulation is generally drier than the NOAH-MP simulation (Figure 4c).

In summary [..[65] ]from the results presented here and in the previous sections, we see that the spatial patterns of land-atmosphere coupling and the climatology of extreme indices [..[66] ]are similar in our WRF simulations (Figures 1 and 2 and S10-S12 in the supplementary information), indicating that the LSM configuration is not influencing these spatial structures. Therefore, other factors common in our four WRF simulations, such as land cover, topography, the latitudinal gradient or atmospheric parameterizations, generate the spatial distribution of the coupling metrics and the extreme indices. Nonetheless, each LSM configuration yields different degree of land-atmosphere coupling and different values of temperature and precipitation extreme events at local scales. Thus, the CLM4 LSM is identified as the component yielding the strongest land control on surface conditions and the highest temperatures during cold and warm events over most of North

---

[65]removed: , although all simulations represent a similar spatial pattern of

[66]removed: , each LSM simulation produces different values for

# CLIMATOLOGY OF EXTREMES

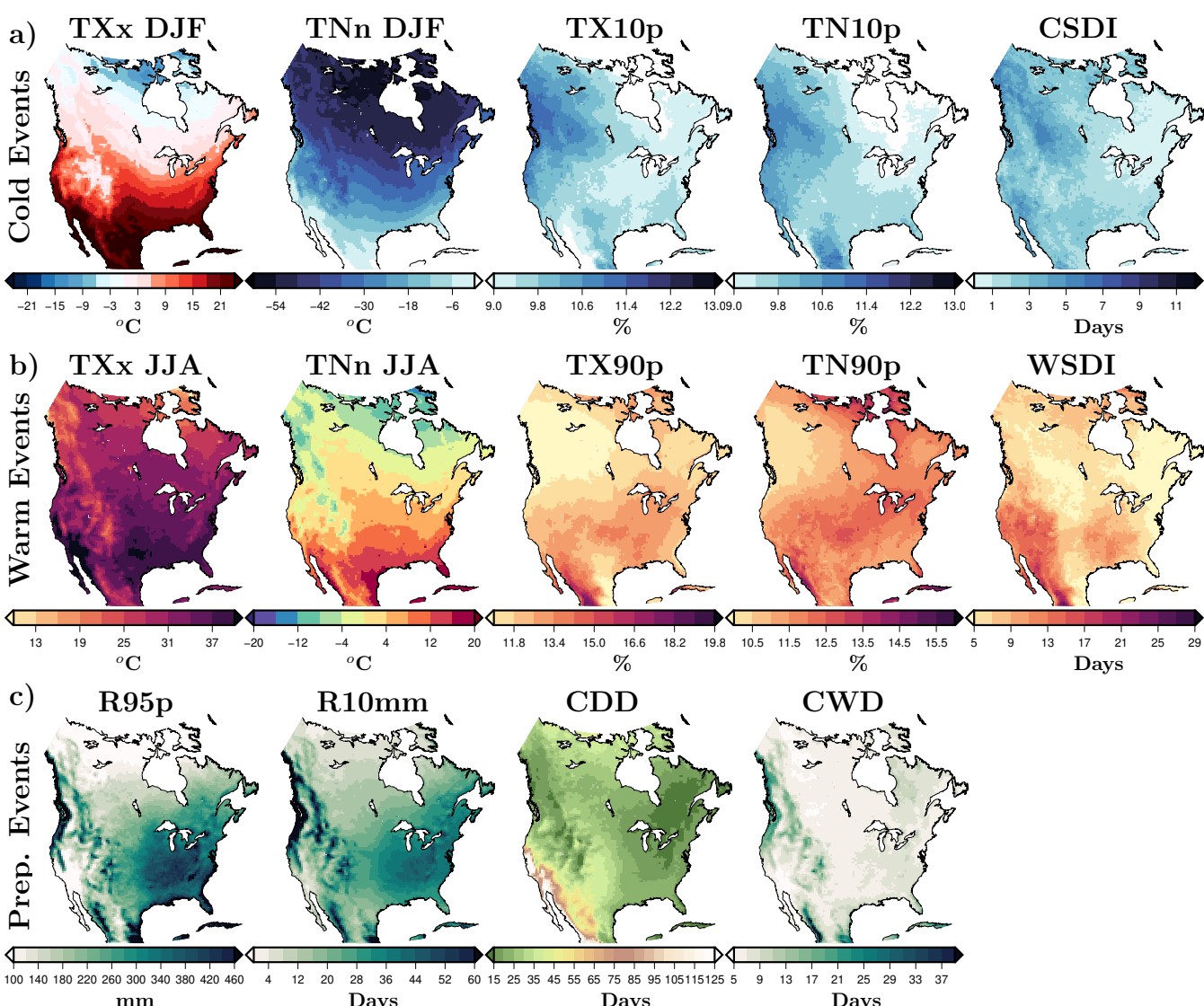

**Figure 3.** Climatology of extreme indices associated with cold temperature events (a), warm temperature events (b), and precipitation events (c) for the [..[57] ]ensemble mean, formed by the four WRF simulations (Table 2: TXx/TNn, maximum/minimum value of the maximum/minimum daily temperatures; TN10p/TX10p, percentage of cold nights/days; TN90p/TX90p, percentage of hot nights/days; CSDI/WSDI, cold/warm spell duration index; R95p, total annual precipitation in wet days; R10mm, number of wet days in a year; CDD/CWD, consecutive dry/wet days). The climatology of each index is estimated as the mean of each extreme index at each grid cell for the analysis period (1980-2012).

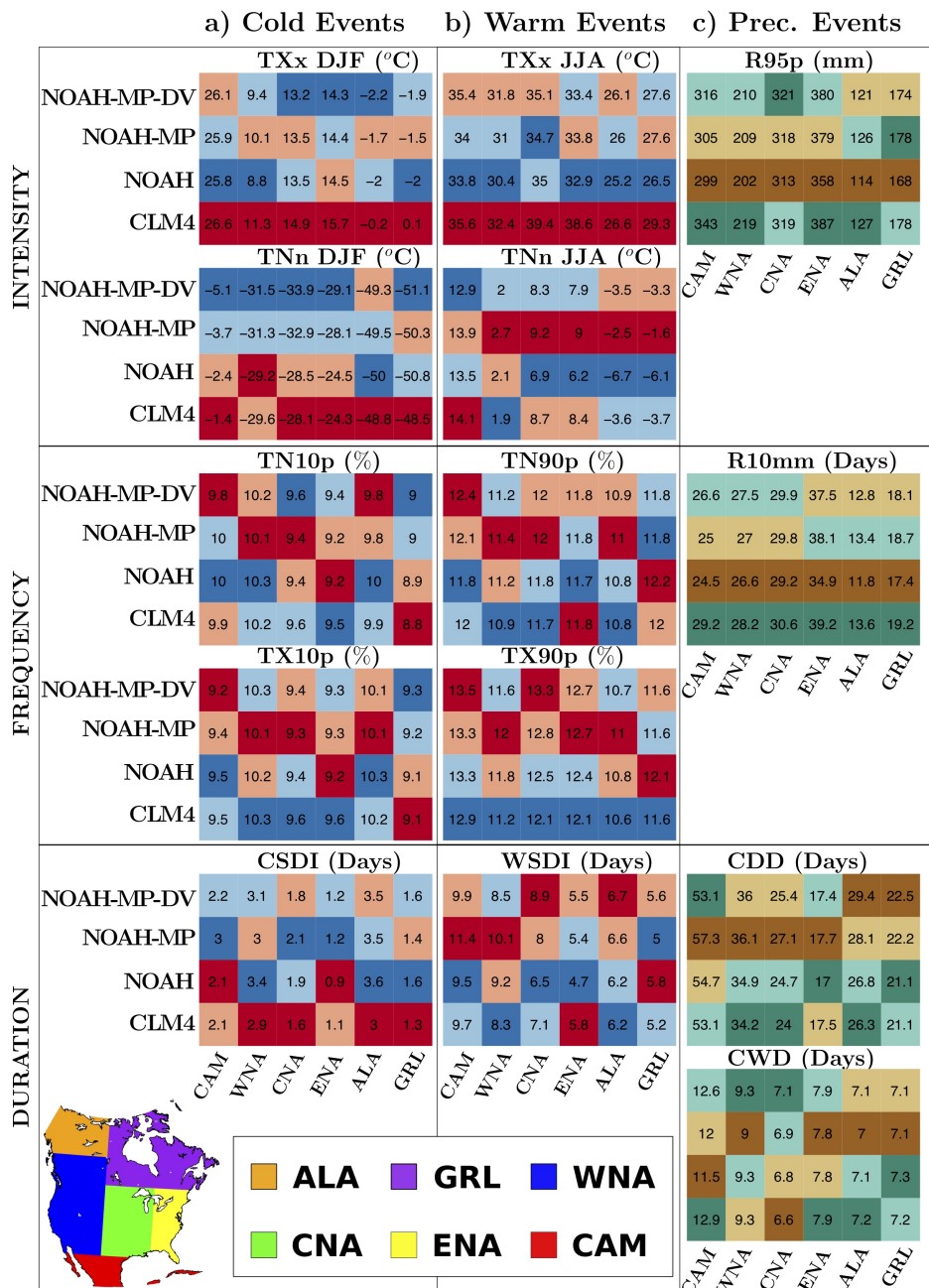

**Figure 4.** Comparison of the simulated climatologies of temperature and precipitation extreme indices included in Table 2 among the WRF simulations averaging over six land North American regions adapted from Giorgi and Francisco (2000) (Central America, CAM; Western North America, WNA; Central North America, CNA; Eastern North America, ENA; Alaska, ALA; and Greenland, GRL). [..[58] ]Colors correspond to the [..[59] ]hottest (red) and coldest (blue) index values among the WRF simulations for the [..[60] ]representation of cold (a) and warm (b) temperature [..[61] ]extremes, and to the driest (brown) and wettest (green) [..[62] ]index values for the [..[63] ]representation of precipitation [..[64] ]extremes (c) over each region.

America as well as the heaviest and most frequent precipitation extremes over most locations [..[67] ](Figures 1, 2 and 4). That is, the simulation with more coincidences of extreme high (low) LH and extreme low (high) SAT is also representing the most intense temperature and precipitation extremes. This suggests that the simulation of very low latent heat flux may be influencing the simulation of heat extremes by inducing an increase in the energy available for sensible heat flux, and likely increasing air temperatures. Meanwhile, the [..[68] ]simulation of high latent heat flux may increase the representation of atmospheric water content, inducing changes in the formation of clouds and precipitation. Thus, the strong land control on the CLM4 simulation seems to enhance the intensity of warm and heavy precipitation events comparing with the rest of simulations, particularly in comparison with the NOAH simulation. The NOAH LSM produces the weakest land control on surface conditions and one of the [..[69] ]lowest intensities for all temperature [..[70] ]indices as well as the lowest intensity and frequency of heavy precipitation events over all regions. The comparison of the NOAH-MP simulations using prescribed and dynamic vegetation shows that the use of dynamic vegetation yields stronger land control at low and middle latitudes in summer and more intense, frequent and longer heavy precipitation events over the same regions (Figures 1, 2 and 4). Thus, this comparison also supports that the simulation of strong land control leads to heavier precipitation events.

### 4.3 LSM uncertainty in the simulation of temperature and precipitation extremes

Although all WRF simulations show similar spatial patterns for temperature and precipitation extreme indices (Figures [..[71] ]S10, S11 and S12), there are large uncertainties in the climatology of each extreme index associated with the use of different LSM [..[72] ]configurations. For the simulation of the intensity of cold events, the multi-model range across the WRF simulations for the hottest day in DJF (TXx DJF) shows large values over the boreal forest and the Rockies, where the index climatology is close to $0^{o}$C (Figures 3 and 5a). The representation of the coldest night in DJF (TNn DJF) shows large LSM dependency, yielding ranges up to 12 $^{\circ}C$ over the US and a spatial average of 4 $^{\circ}C$, displaying large uncertainties over areas where the index climatology approaches to $0^{o}$C (Figures 3 and 5a). The simulated intensity of warm temperature events, measured by the temporal average of the hottest day in summer (TXx JJA), differs up to 10 $^{\circ}C$ among simulations over eastern North America, with a spatial average of 3.5 $^{\circ}C$ (Figure 5a). The simulation of the mean coldest night in summer (TNn JJA) varies across simulations from 2 to 3 $^{\circ}C$ over the whole domain, except in the Arctic where the range across simulations reaches approximately 15 $^{\circ}C$ and the index value yields negative temperatures for some simulations (Figure 3 and 5a). The frequency of warm extreme temperature events varies among simulations; the range for the number of hot days (TX90p, based on maximum temperatures) is up to $4.2\%$ over the US with a spatial average of $0.97\%$ over the whole domain, and the range for the number of hot nights (TN90p, based on minimum temperatures) reaches values up to $3.8\%$ at low latitudes with a spatial average of approx. $0.7\%$ (Figure 5b). Large values of the multi-model range for the number of hot days (TX90p[..[73] ]) approximately

---

[67]removed: .

[68]removed: NOAH LSM produces

[69]removed: weakest climatologies

[70]removed: and precipitation extreme indices over most regions.

[71]removed: S5, S6 and S7

[72]removed: components

[73]removed: index

coincide with the largest index values (Figures 3 and 5b). Note that ranges of more than $2\%$ in the number of hot days and nights correspond to differences of more than 7 days per year in the index climatology simulated by different LSMs. Ranges of indices related to the frequency of cold events show smaller values than those for warm temperature events, displaying no clear spatial pattern with averages of $\sim 0.5\%$ (i.e. 1.8 days per year) for the number of cold days and nights (TX10p and TN10p; Figure 5b). The duration of warm spells is greatly affected by the choice of the LSM component, while its effect is weaker on the simulated duration of cold events (Figure 5c). The range of the duration of warm spells across simulations yields values of more than 10 days over Mexico and over broad areas of the central and southern US, with a spatial average of 2.8 days (Figure 5c). Otherwise, the LSM effect on the simulated duration of cold spells is weaker, reaching differences of about 6 days among simulations in central Canada with a spatial average of 1.3 days (Figure 5c). For both indices, the LSM differences are larger where the duration indices display larger values (Figure 3 and 5c).

The simulated climatology of the intensity of extreme precipitation events is also strongly affected by the [..[76] ]configuration of LSM, with the total annual precipitation in wet days (R95p[..[77] ]) reaching LSM differences larger than 100 mm at low latitudes and over the eastern US with a spatial average of 39 mm (Figure 6a). The frequency of heavy precipitation events varies among simulations [..[78] ]by about 35 days per year at some locations in Mexico and the US, with a spatially averaged range of 3.5 days per year (Figure 6b). The areas with the largest inter-model range of the precipitation frequency index across simulations are located in Mexico, the Rockies and at some grid cells over the eastern US coast (Figure 6b). The simulation of the number of consecutive dry and wet days also depends on the choice of the LSM component, presenting larger differences among simulations in the climatology of the consecutive dry days index than in the climatology of the consecutive wet days index (Figure 6c). The inter-model range across LSM simulations reaches 37 days for the number of consecutive dry days over central and southwestern North America, with a spatial average of 4 days per year (Figure 6c). Meanwhile, the simulated number of consecutive wet days also shows LSM differences of more than 20 days at a few grid cells, but lower values over most of the domain, yielding a spatial average of $\sim 1.2$ days (Figure 6c). Large inter-model ranges of precipitation indices across WRF simulations coincide with areas where each index reaches the maximum values (Figure 3 and 6).

Results for the VAC metric present some similarities with the spatial pattern of uncertainties in the WRF simulation of temperature and precipitation extreme events[..[79] ], [..[80] ]which suggest a relationship between these results. The areas showing large uncertainty in the simulation of the intensity indices of cold extremes coincide with areas where LSM simulations differ in the representation of DJF atmospheric control VAC categories (VAC$_a$ and VAC$_b$; Figures 1 and 5). [..[81] ]Particularly,

---

[76]removed: choice

[77]removed: index

[78]removed: in

[79]removed: . We estimate the seasonal component of each index to compare with the metrics of land-atmosphere coupling seasonally, except for the temperature intensity indices which are defined in JJA and DJF (Figures S8

[80]removed: S9, and S10)

[81]removed: The seasonal components of the inter-model range for the simulated percentage of cold days and nights show small uncertainty in winter with very noisy fields (Figure S8). However, the seasonal decomposition of the range for the duration index of cold events shows a region with large uncertainty over western NA in MAM, corresponding to an area with marked differences between LSM simulations in the MAM atmospheric control VAC categories (Figures S8c and 1)

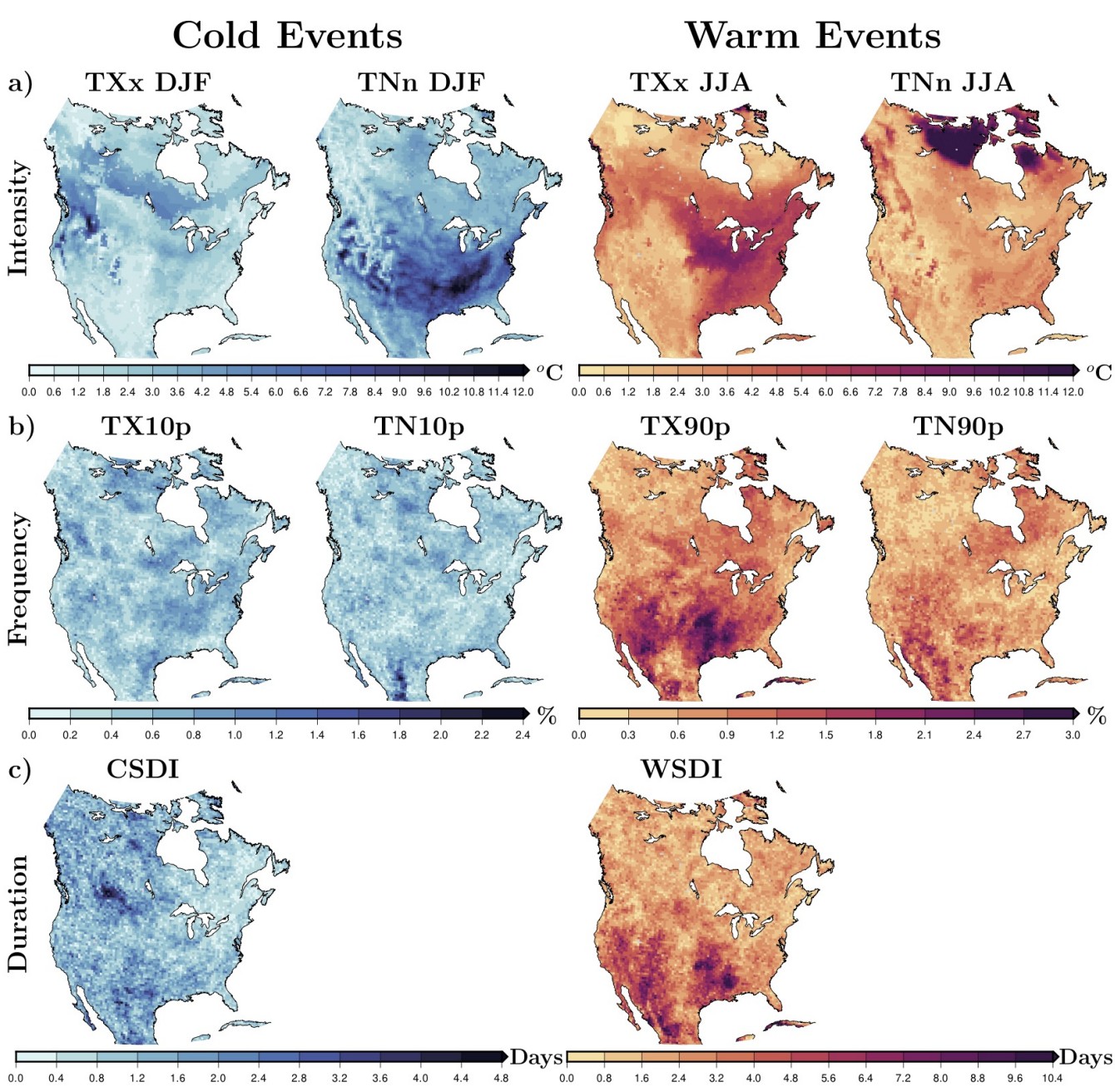

**Figure 5.** Multi-model ranges across the WRF simulations (i.e., difference between the highest value and the lowest value of the [..[74] ]four WRF simulations at each grid cell) of extreme indices associated with the intensity (a), frequency (b), and duration (c) of cold (left) and warm (right) extreme temperature events ([..[75] ]TXx/TNn, maximum/minimum value of the maximum/minimum daily temperatures; TN10p/TX10p, percentage of cold nights/days; TN90p/TX90p, percentage of hot nights/days; CSDI/WSDI, cold/warm spell duration index). The range among simulations is computed using the mean of each index from 1980 to 2012 for each simulation.

# Precipitation Events

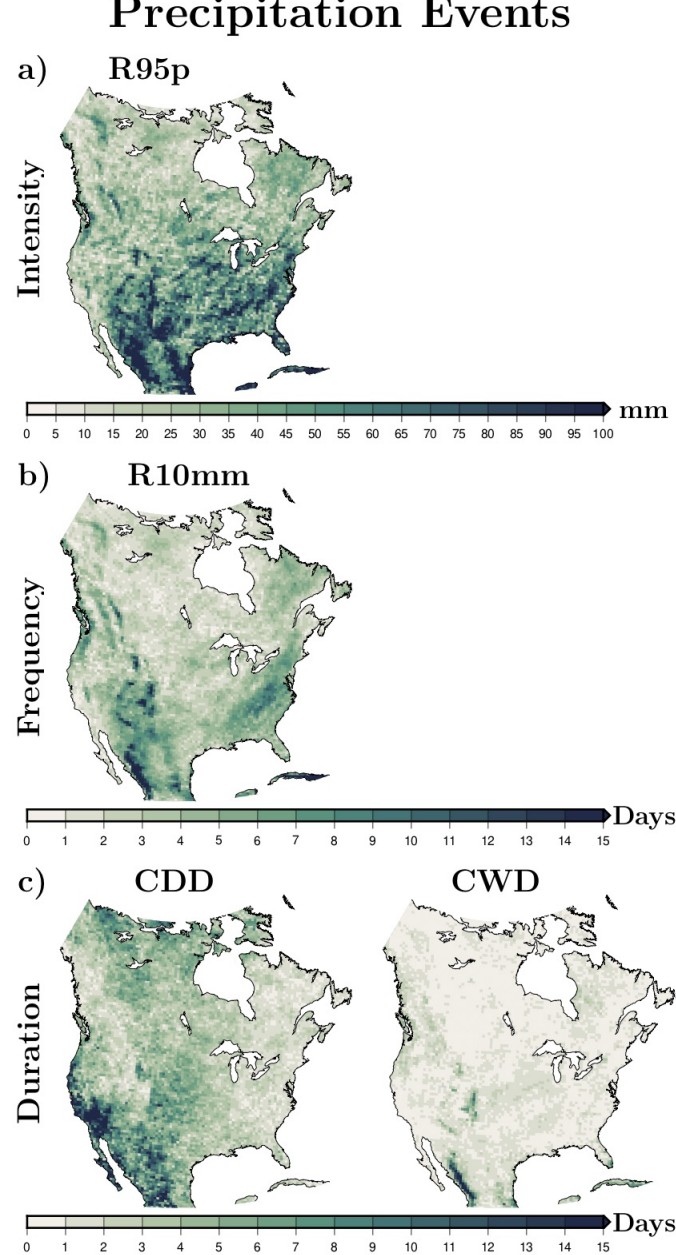

**Figure 6.** As in Figure 5 but for extreme precipitation events (R95p, total annual precipitation in wet days; R10mm, number of wet days in a year; CDD/CWD, consecutive dry/wet days).

the uncertainty in the hottest day in winter is larger over areas with evergreen needleleaf forest (Figure 5 and S1 in the supplementary information). Thus, although all simulations include the same land use categories, the differences in the representation of vegetation by each LSM (Figure S13 in the supplementary information) from the plant functional types used by the CLM4 LSM to the canopy cover simulated by the NOAH LSM are likely related to the differences in the simulation of land-atmosphere coupling and extreme indices. For the simulation of warm extremes, large LSM differences in the intensity indices correspond to LSM differences in the JJA VAC categories associated with [..[82] ]the energy-limited areas (Figures 1 and 5). [..[83] ]The areas with large uncertainty in the [..[84] ]hottest day in summer also correspond with areas showing [..[85] ]a mix of vegetation from croplands to forests (Figure S1 in the supplementary information). Thus, these results also suggest that LSM differences in the representation of vegetation cover are causing a different representation of land-atmosphere interactions in energy-limited areas, and consequently different climatologies of the hottest day among our simulations. The uncertainty in the simulation of the coldest night in summer is larger in areas over the mixed tundra category, where LSM configurations differ in the simulation of snow cover in summer (Figures S1 and S13 in the supplementary information). Thus, LSM differences in the representation of snow cover from the single snow layer simulated by the NOAH LSM to the five layers simulated by the CLM4 LSM may also contribute to the uncertainty in the intensity index of warm [..[86] ]events. The uncertainty in the number of hot days and the duration of warm spells is larger over regions under land control[..[87] ], particularly over open shrub-lands, suggesting the possible influence of LSM differences in the simulation of soil moisture (Figures 2 and S1 in the supplementary information). The range of the intensity index of precipitation extremes displays a large JJA component over areas under land control at low latitudes and under atmospheric control at middle and high latitudes (Figures [..[88] ]1 and 2 [..[89] ]and S14a). For the intensity index of heavy precipitation events, our simulations show large uncertainties in areas with mixed vegetation (Figure 6 and S1 in the supplementary information), suggesting the influence of LSM differences in the representation of vegetation cover on the simulation of latent heat flux, thus leading to changes in the simulation of atmospheric water content and precipitation. The uncertainty in the intensity, frequency and duration of heavy precipitation events is high over the western Mexican coast, where the model is representing the tropical forest and the NOAH simulation showed strong atmospheric control in disagreement with the rest of our simulations (Figures 1 and 6 and S1 in the supplementary information). Thus, these results suggest that LSM differences in the description of vegetation and snow cover (e.g the number of snow layers and the description of the canopy) are leading to uncertainties in the simulations of precipitation extremes. Although our results suggest that

---

[82]removed: atmospheric control episodes

[83]removed: Areas

[84]removed: JJA simulation of warm frequency indices coincide

[85]removed: strong land control on surface processes as well as regional differences between LSM simulations (Figures S9ab and 2). The duration

[86]removed: extremes also shows large inter-model range in JJA

[87]removed: (Figures S9c and 2

[88]removed: S10a,

[89]removed: ). The MAM and SON components of the range for the precipitation intensity index also show large values over areas with atmospheric control in MAM and over areas with land control in SON (Figures S10a, 1 and 2). The frequency index of precipitation events presents large inter-model range over small regions in JJA, coinciding with areas under atmospheric control (Figures S10b and 1 ). The inter-model range of dry periods coincides with land control areas at low latitudes in all seasons and with atmospheric control areas at high latitudes in MAM (Figures S10c 1 and 2) . The inter-model range

the LSM representation of land cover can be related to the uncertainty in the simulation of [..90 ]extreme indices, the differences in the VAC metric and in the extreme indices are larger between different LSM components than those between simulations with prescribed and dynamic vegetation (Figures 1, 2 and 4).

In order to address the LSM influence on the simulation of extreme events, we compute the ranges among our four WRF simulations using the 95th percentile of the analysis period for each extreme index. The uncertainty in the WRF simulations due to the LSM component when using the 95th percentile for each extreme index leads to similar conclusions (Figures [..91 ]S15 and S16 in the supplementary information). The LSM differences using the 95th percentile of the analysis period are larger for all extreme temperature and precipitation indices than using the period mean as expected, but the marked areas are analogous (Figures 5, 6, [..92 ]S15 and S16). The agreement in the representation of areas with large uncertainty in extreme indices between results using mean and extreme climatologies suggests the LSM influence on extreme events at climatological and shorter time scales.

## 4.4 Comparison between WRF simulations and three CORDEX Evaluation simulations

The climatologies of temperature and precipitation extreme statistics as simulated by the three RCMs participating in the NA-CORDEX project (Table [..93 ]3) show similar spatial patterns to [..94 ]our four WRF simulations (Figures S10-S12 and S17-S19). These similarities in the spatial pattern of extreme indices represented by WRF and the CORDEX RCMs further support the hypothesis that the spatial features of these maps are controlled by topography, land cover and the latitudinal gradient, since the CORDEX RCMs employed atmospheric models and boundary conditions different to our WRF simulations. Although the spatial patterns are similar in both ensembles, the WRF simulations yield colder minimum temperatures in DJF (TNn DJF) and less frequent cold nights (TX10p) than the CORDEX simulations (Figures [..95 ]S10 and S17). The percentage of hot days, however, is higher and warm spells are longer in the WRF simulations than in the CORDEX simulations, particularly over southwestern NA (Figures [..96 ]S11 and S18). The intensity of heavy precipitation extremes is generally higher within the WRF ensemble than in the CORDEX ensemble, while dry periods are longer in the CORDEX simulations (Figures [..97 ]S12 and S19).

The uncertainties in the simulation of extreme statistics within the CORDEX ensemble show some similarities with the WRF uncertainties [..99 ]which arise from the LSM [..100 ]configuration. For example, the simulated climatology of DJF coldest night (TNn DJF) shows large uncertainties over the US for both ensembles, particularly over the eastern US (Figures 5a and 7a).

---

[90]removed: consecutive wet days is large in JJA over a small Mexican region classified under atmospheric control with different degree of coupling between simulations (Figures S10d and 1

[91]removed: S11 and S12

[92]removed: S11 and S12

[93]removed: S1

[94]removed: those from the WRF ensemble (Figures S5-S7 and S13-S15). Although

[95]removed: S5 and S13

[96]removed: S6 and S14

[97]removed: S7 and S15

[99]removed: arising

[100]removed: component

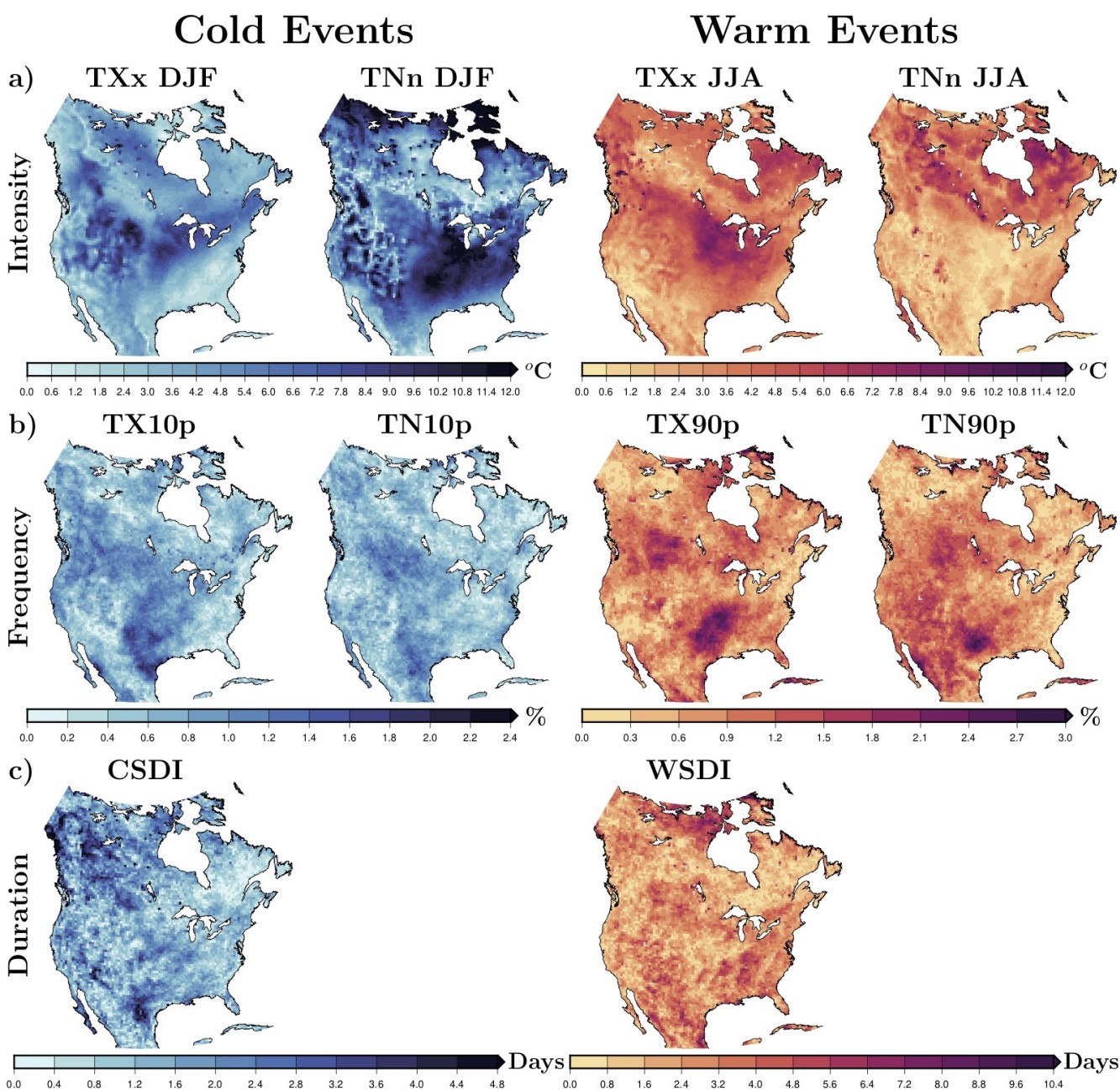

**Figure 7.** Inter-model range across three CORDEX simulations (i.e., difference between the highest value and the lowest value of the three CORDEX [..[98] ]simulations at each grid cell) of extreme indices associated with intensity (a), frequency (b), and duration (c) of cold and warm extreme temperature events (Table 2). The range across simulations is computed using the mean of each index from 1980 to 2012 for each simulation.

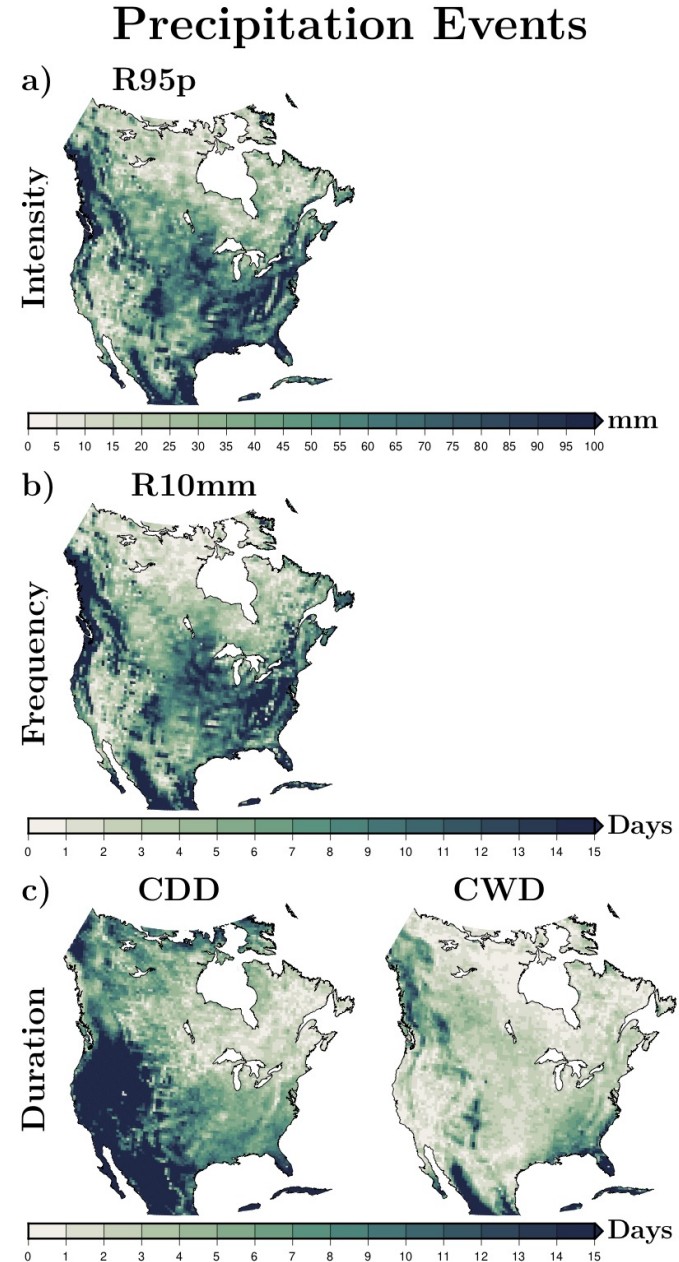

**Figure 8.** As in Figure 7 but for extreme precipitation events.

The climatologies of DJF hottest day (TXx DJF) display large inter-model range within the WRF ensemble over areas where temperatures approximate to $0^oC$, expanding southward for the CORDEX ensemble. The CORDEX inter-model ranges of the frequency indices for cold extremes do not show a clear spatial pattern in agreement with the WRF ensemble. There is, however, a region over the central US with slightly larger ranges among the CORDEX simulations than among the WRF simulations (Figures 5b [..[101] ]and 7b and S20 in the supplementary information). The duration of cold spells presents large uncertainties in the CORDEX ensemble over the eastern US/Mexican border and over western Canada, coinciding with a small region with large inter-model range among the WRF simulations (Figures 5c and 7c). For the simulation of warm temperature extremes, the uncertainties in the intensity indices among the CORDEX simulations show large ranges over the eastern US for the JJA hottest day (TXx JJA) in agreement with the WRF simulations, and at high latitudes for the coldest night (TNn JJA), including the eastern region of Hudson Bay also marked by the WRF ensemble (Figures 5a and 7a). The frequency indices of warm events show large inter-model range across the CORDEX simulations over the central US, also shown in the WRF simulations for the TX90p index (Figures 5b and 7b). The uncertainty in the duration of warm spells among the CORDEX simulations does not show large spatial differences, although the ranges are slightly larger at low latitudes coinciding with regions marked by the WRF ensemble and at very high latitudes (Figures 5c and 7c). The simulation of precipitation extreme statistics is generally more uncertain across the CORDEX simulations than across the WRF simulations (Figures 6, 8, and [..[102] ]S21 in the supplementary information). Interestingly, all regions with large uncertainties in the simulation of precipitation extremes among the WRF simulations are also identified as areas with large uncertainty across the CORDEX ensemble. There are, however, additional areas with large uncertainty in the CORDEX ensemble, particularly for the consecutive dry days index and the frequency index at middle and high latitudes (Figures 6 and 8). [..[103] ]The larger spread of the precipitation indices within the CORDEX ensemble in comparison with the spread in our WRF simulations (Figure S21 in the supplementary information) was expected due to the use of different atmospheric models in the CORDEX ensemble. Nonetheless, the agreement between the WRF and CORDEX [..[104] ]simulations in the placement of areas with large uncertainties suggests that results from this study may be applicable to other [..[105] ]modelling experiments, particularly for the simulation of warm temperature and precipitation extremes.

## 5 Discussion

### 5.1 Comparison of inter-model ranges across the WRF and CORDEX ensembles

In order to provide context for the applicability of these results to other [..[106] ]sets of simulations, we compared the [..[107] ]range across our WRF simulations with the inter-model range across three CORDEX simulations in representing extreme

---

[101] removed: , 7b , and S16b

[102] removed: S17

[103] removed: Thus, the comparison

[104] removed: ensembles

[105] removed: model ensembles over some areas

[106] removed: model ensembles

[107] removed: inter-model range across the

events (Figures 5-8). Since CORDEX simulations were performed by three structurally different RCMs (the WRF, the RCA4, and the CRCM-UQAM models), we expected a broader inter-model range of the simulated extreme indices across CORDEX simulations. Differences in the representation of extreme events among the CORDEX simulations arise from several factors, such as different atmospheric and ocean parameterizations, land surface model components, the representation of land cover, treatment of boundary conditions and the application of nudging techniques. In addition to all these factors, [..108 ]the sensitivity

to initial conditions in models may be another important [..109 ]factor for the inter-model range of the simulated extreme events. The WRF sensitivity to initial conditions may also affect the interpretation of the differences among our four simulations with different LSM configurations. However, previous analyses using the WRF model (Liu et al., 2019; Gallus and Bresch, 2006) as well as other climate models (Kharin et al., 2007; de Elía et al., 2008; Sillmann et al., 2013a) have showed that the spread of extreme events among ensemble members of an individual model is generally small compared to inter-model

spreads [..110 ]or the differences arising from different physics configurations.

Although the CORDEX simulations were performed using boundary conditions from the ERA reanalysis product, the comparison with the WRF simulations is possible because we compute [..111 ]the ranges across simulations as a measure of the uncertainty in each [..112 ]simulation ensemble. Thus, we compare [..113 ]the uncertainty in each set of simulations finding common areas with large [..114 ]ranges for the [..115 ]representation of cold and warm temperature extremes and precipitation

extremes, despite the fact that they used different products as boundary conditions. The [..116 ]agreement in the placement of areas with large uncertainties in the representation of extreme events [..117 ]within the CORDEX ensemble [..118 ]and those within our WRF simulations suggests that the uncertainties in these areas may arise from similar causes. Our WRF simulations only differ in the configuration of the LSM component. Therefore, the differences between LSM components can also be an important source of uncertainty in the [..119 ]simulation of extreme events within the CORDEX simulations,

through a different representation of land-atmosphere [..120 ]interactions.

One of the simulations included in the CORDEX ensemble was performed by the WRF model using the NOAH LSM component. The comparison of the extreme indices between our WRF-NOAH simulation and the one included in the CORDEX ensemble shows similar spatial patterns and regional differences in the value of each extreme index (second column in Figures S11-S13 and third column in Figures S17-S19 in the supplementary information). However, this com-

parison is not very different if we use another CORDEX simulation performed by a different RCM. This suggests that

---

[108]removed: internal variability

[109]removed: component

[110]removed: (Kharin et al., 2007; Sillmann et al., 2013a)

[111]removed: inter-model ranges across ensembles

[112]removed: model

[113]removed: model's uncertainty in both ensembles

[114]removed: inter-model

[115]removed: simulation

[116]removed: similar uncertainties

[117]removed: in

[118]removed: relative to the WRF simulations suggest that the LSM component may

[119]removed: CORDEX ensemble. That is, the LSM component employed in each CORDEX simulation (Table S1)may be simulating different

[120]removed: interactions and affecting the simulation of extreme events over those regions

the spatial pattern of the extreme indices is driven by factors common in all simulations, such as land cover, topography and the latitudinal gradient. The regional differences in the value of extreme indices between our WRF-NOAH simulation and the WRF-NOAH CORDEX simulation are likely caused by the use of nudging techniques to match the ERA-Interim product in the CORDEX simulation.

[..[121] ]Although there are more sources of uncertainty in the CORDEX simulations than across the WRF simulations, the comparison between [..[122] ]the uncertainty within each set of simulations (i.e. the difference between the range among the WRF simulations and the range among the CORDEX simulations) displays larger ranges across the WRF simulations than across the CORDEX ensemble over certain areas and for certain extreme indices (Figures [..[123] ]S20 and S21 in the supplementary information). This suggests the possible existence of bias compensation inside the CORDEX simulations.

Moreover, each RCM may have a different sensitivity to the employed LSM component as well as to other components and parameterizations. Additional sensitivity studies using the WRF model or another climate model with different settings and parameterizations may help to discern other important sources of uncertainties in the simulation of extreme events, such as horizontal resolution.

## 5.2   Climatology of extreme events as represented by the WRF simulations and by the CMIP5 simulations

Sillmann et al. (2013a) presented an evaluation of the CMIP5 models in simulating some of the extreme indices defined by ETCCDI; this information was used in the Intergovernmental Panel on Climate Change (IPCC) chapter on models' evaluation (Flato et al., 2013). The analysis period employed by Sillmann et al. (2013a), 1981-2000, differs from the one used in this analysis, but a rough comparison can be done between our results and theirs for some extreme indices. For example, the spatial patterns of DJF coldest night and JJA hottest day are similar for the WRF and CMIP5 ensemble means (Figure 3 and

Figure 2 in Sillmann et al. 2013a). The similarities in the spatial pattern of extreme indices between our WRF simulations, the CMIP5 and the CORDEX ensembles suggest that the topography, land cover and latitudinal gradient are driving these spatial features. Sillmann et al. (2013a) also provides regional averages over six NA regions, adapted from Giorgi and Francisco (2000). These spatial averages allow identification of some regional differences between the WRF and the CMIP5 ensembles, for example over the eastern US coast (ENA region) where the WRF simulations yield warmer JJA maximum

temperatures than the CMIP5 ensemble (Figure 4 and Figure 3 in Sillmann et al. 2013a). The spatial patterns of the WRF and CMIP5 ensembles for CSDI and WSDI indices are also similar, although the WRF ensemble reaches longer cold and warm events (Figure 3 and Figures S6-S7 in Sillmann et al. 2013a). The representation of the intensity index for heavy precipitation events (R95p) also shows similar spatial patterns between both ensemble means, although the WRF ensemble is generally more intense over most regions (Figures 3 and 4, and Figures 6 and 7 in Sillmann et al. 2013a). Similar results are found

for the simulation of consecutive dry days, showing similar spatial patterns with some regional differences especially at low latitudes (CAM region, Figures 3 and 4, and Figures 6 and 7 in Sillmann et al. 2013a). The variability across the CMIP5

---

[121] removed: Despite there being

[122] removed: these ensembles displays larger inter-model

[123] removed: S16 and S17

ensemble for the simulation of precipitation indices seems to be particularly large at low latitudes (CAM region) similar to WRF uncertainty in the representation of precipitation extremes associated with the LSM component (Figure 6, and Figure 7 in Sillmann et al. 2013a). Although this is a rough comparison between results presented in this article and in Sillmann et al. (2013a), this comparison suggests that our conclusions could be also applicable to the CMIP5 ensemble as it was the case for the CORDEX ensemble.

## 5.3 Implications of these results

Increases in heat-related events have been directly and robustly associated with increases in mortality, for example in Europe during the heatwave of 2003 (Fischer et al., 2007) or in India [..[124] ]during the heatwave of 2015 (Pattanaik et al., 2017). Heavy precipitation events often lead to floods, which also are directly associated to economic loss and death toll (Hu et al., 2018). All climate change projections point [..[125] ]to a future increase in temperature and precipitation extreme events (Sillmann et al., 2013b), thus developing mitigation strategies will become necessary to preserve human health. Climate model simulations are our best source of information to [..[126] ]mitigate climate change impacts. However, the results presented here indicate that the simulation of several extreme indices varies largely depending on the employed LSM component, because of the different representation of land-atmosphere interactions. This means that a climate model may simulate the climatology of heat extremes $5^{o}C$ warmer and 6 days longer depending on the employed LSM component, and similarly for cold extremes and heavy precipitation events. Therefore, studies based on multi-model ensembles and reanalyses should include a variety of LSM configurations to account for the uncertainty arising from this model component or to test the performance of the selected LSM component before performing the whole simulation. The accuracy of climate models and the management of uncertainties in simulating extreme events will likely affect climate change policy, therefore having repercussions for society and environment.

The indices employed here to study the climatology of extreme temperature events were based on minimum and maximum temperature outputs. However, many studies have proven that the study of compound events using indices based on multiple variables, such as temperature and moisture outputs, are more representative of thermal stress in humans and ecosystems than standard indices (Zscheischler et al., 2018). The large LSM influence on the climatology of extreme temperature and precipitation events, suggests that the uncertainty arising from the LSM component could be higher on extreme indices based on multiple variables. However, the analysis of the LSM influence on compound events is beyond the scope of this work, and constitutes an interesting line for future research.

---

[124]removed: (Mazdiyasni et al., 2017)

[125]removed: out

[126]removed: inform measure against

## 6 Conclusions

WRF simulations over North America coupled to different LSM components showed similar spatial patterns of land-atmosphere interactions [..[127] ]as measured by the Vegetation-Atmosphere Coupling (VAC) index. The use of this metric allows the classification of our results into: energy-limited areas, where atmospheric conditions control land-atmosphere interactions ($VAC_a$ and $VAC_b$); and water-limited areas, where soil moisture deficits control the energy and water exchanges between the land surface and the lower atmosphere ($VAC_c$ and $VAC_d$ categories). Our results indicate atmospheric control over land-atmosphere interactions at middle and high latitudes and land surface control over lower latitudes, particularly

in JJA. However, the simulation of land-atmosphere [..[128] ]coupling differs at regional scales depending on the LSM choice in two directions; by altering land control on surface processes ($VAC_c$ and $VAC_d$ categories) and by altering atmospheric [..[129] ]conditions and its influence on [..[130] ]land-atmosphere interactions ($VAC_a$ and $VAC_b$ categories). Thus, the NOAH LSM is associated with the weakest representation of land control on surface conditions, while the CLM4 LSM simulates one of the

strongest land [..[131] ]effects on surface conditions. The use of different LSM components leads to large ranges of represented extreme temperature and precipitation events, affecting their simulation in intensity, frequency and duration. The CLM4 LSM yields the weakest cold events, the warmest hot days, and the heaviest precipitation events, while the NOAH simulation yields the weakest [..[132] ]warm temperature events and the weakest heavy precipitation events. [..[133] ]Meanwhile, the NOAH-MP LSM produces the driest simulation, yielding slightly wetter conditions when using dynamic vegetation at middle and low

latitudes. [..[134] ]Although the LSM differences in our results are more marked than differences between the simulations with prescribed and dynamic vegetation, [..[135] ]the use of dynamic vegetation yields stronger land control at low and middle latitudes in summer and more intense, frequent and longer heavy precipitation events and reduces the duration of droughts over the same regions. Thus, our results suggest a relationship between the degree of land control on surface conditions reached by each LSM configuration and the intensity of extreme events, in agreement with the case study

during the Russian 2010 heat wave (Zscheischler et al., 2015).

  Previous studies using GCM simulations suggested a dependence of the simulated land-atmosphere interactions on the employed LSM component with possible consequences for the simulation of extreme events (García-García et al., 2019). Results from four WRF simulations differing only in the LSM [..[136] ]configuration support that hypothesis, identifying LSM differences in the description of land cover as a possible cause. Additionally, areas with large uncertainties in the simulation

of temperature and precipitation extremes across the WRF simulations due to different LSM components appear in the NA-

---

[127]removed: , indicative of atmospheric control over surface conditions

[128]removed: interactions

[129]removed: forcing

[130]removed: surface conditions

[131]removed: effect

[132]removed: land control on surface conditions, the weakest

[133]removed: This relationship between the degree of land control on surface conditions and the intensity of extreme events is in agreement with two case studies during the Russian 2010 heat wave and the Amazon 2010 drought (Zscheischler et al., 2015).

[134]removed: Despite small differences between

[135]removed: differences are much more marked among the WRF simulations due to different LSM components

[136]removed: component

CORDEX model ensemble, which indicates the possible LSM influence on the simulation of extreme events within other model ensemble. This work reinforces the important role of the LSM component in climate simulations, supporting the urgency of on-going research focused on improving this model component and their implementation in regional and global climate models as well as in reanalysis products. The strong LSM dependency of climate model simulation of extremes is also of special importance for international reports focused on land, such as the IPCC Special Report on Climate Change, Desertification, Land Degradation, Sustainable Land Management, Food Security, and Greenhouse gas fluxes in Terrestrial Ecosystems (Arneth, 2019). Future sensitivity analyses to the LSM component using different regional and global climate models would be useful to understand models' differences in simulating temperature and precipitation extremes, helping to narrow the inter-model range across reanalyses and climate model projections in simulating extreme events.

*Code and data availability.* The source code of the Weather Research and Forecasting model (WRF v.3.9) is http://www2.mmm.ucar.edu/ wrf/users/download/get_source.html (access date: August, 2017). Extreme indices were calculated using ETCCDI definitions (https://www. climdex.org/learn/indices/) and software from the R package climdex.pcic (http://cran.r-project.org/web/packages/climdex.pcic/index.html, November 2019). The daily temperature and precipitation outputs of all simulations together with the code used to estimate the presented climate extreme indices are available at https://doi.org/10.5281/zenodo.3745510. The NARR product was obtained from https://nomads. ncdc.noaa.gov/data/narr/ (access date: August, 2017). Evaluation simulations from the North American component of the CORDEX project were downloaded from https://www.earthsystemgrid.org/search/cordexsearch.html (access date: December, 2018).

*Author contributions.* AGG designed the modeling experiment, performed the simulations and analyzed model outputs. All authors contributed to the interpretation and discussion of results. AGG wrote the manuscript with continuous feedback from all authors.

*Competing interests.* The authors declare that they have no conflict of interest

*Acknowledgements.* We thank to the Mesoscale and Microscale Meteorology (MMM), the National Center for atmospheric Research (NCAR), the National Oceanic and Atmospheric Administration (NOAA) for making the WRF code and the NARR product available as well as to the World Climate Research Program's Working Group on Coupled modeling, which oversees CORDEX, and the individual groups and institutions for making their data available. This work was supported by grants from the Natural Sciences and Engineering Research Council of Canada Discovery Grant (NSERC DG 140576948), the Nova Scotia Research and Innovation Trust (NSRIT), Compute Canada and The Canadian Foundation for Innovation (CFI) to H. Beltrami. H. Beltrami holds a Canada Research Chair (CRC 230687). A. García-García and F.J. Cuesta-Valero are partially financed by Dr. Beltrami's CRC and Memorial University of Newfoundland.

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
