# Peer review of "Land Surface Model influence on the simulated climatologies of temperature and precipitation extremes in the WRF v.3.9 model over North America"

_Geoscientific Model Development, 2020_

## Referee Comment (RC1) · Anonymous Referee #1 · 28 May 2020

Review for gmd-2020-86: "Land Surface Model influence on the simulated climatologies of temperature and precipitation extremes in the WRF v.3.9 model over North America" by García-García et al.

In their article "Land Surface Model influence on the simulated climatologies of temperature and precipitation extremes in the WRF v.3.9 model over North America", García-García et al. use 4 different land surface model (LSM) configurations coupled to the WRF regional atmospheric model over a regional North America domain, to explore the sensitivity of precipitation and temperature extremes to the choice of land model. They find similar overall patterns in the strength of land-atmosphere coupling across their simulations but show that the strength of that coupling can differ significantly across LSMs.

This study is a clear demonstration of how the uncertainty in temperature and precipitation is highly dependent on the choice of land model – a subject too-often overlooked in the study of extremes. The authors provide a valuable contribution to the role of the choice of LSM in the analysis of land-atmosphere coupling and modeling extreme events.

Prior to publication, this study would benefit from elaboration on how the authors' results would vary had they explored multiple instantiations of the same LSM-WRF coupling, rather than a single instance of each LSM. Portions of the text were at times dense and confusing – a reworking of these sections (pointed out below) would greatly benefit the reader. Lastly, I would like to see a deeper analysis of the mechanisms driving the differences between the LSM results. The authors may consider this beyond the scope of their study, but explaining why the LSMs generate different results would be extremely useful to the community, rather than simply pointing out that different LSMs result in different distributions of extremes in temperature and precipitation. Following revision, this manuscript would be appropriate for publication in GMD.

**Specific Comments:**

**Major:**

- The authors refer to an "ensemble of simulations". To clarify, they did not actually run an ensemble for each model setup, correct? Rather, there are 4 simulations in total evaluated: NOAH/WRF, NOAH-MP/WRF, NOAH-MP-DV/WRF, and CLM/WRF? The authors should clarify the text on this point.
- Related to (and more important than) the previous comment, the authors do not discuss what variability within a single LSM-WRF framework is expected. Had the authors instead ran an ensemble of, say, 10 NOAH/WRF simulations with slightly different initial conditions, how large a spread in the strength of land-atmosphere coupling and the statistics associated with extreme events would they see? Is the spread we're seeing across the 3/4 LSMs simply due to the fact that WRF was run 4 times, or is it truly a physical response to the physics and mechanisms of the particular LSMs?
- A discussion of *why* the land-atmosphere coupling strength varies between their simulations would not only help show the differences between the 4 LSM/WRF simulations are actually the result of structural differences in the land model, but would

also be greatly beneficial to the reader for understanding why they should care about the spread in LSMs. Including a discussion on this topic would also help the reader select an LSM that appropriately represents the aspect of land-atmosphere coupling they may be interested in studying. I don't mean that the authors need to completely restructure the paper to address this topic. Rather, they often make statements like (Line 203, just an example) "LSM differences in the representation of VACa and VACb probabilities suggest the LSM influence on the evolution of atmospheric conditions". Rather than simply reporting differences in VACa-d, it would be useful for the authors to elaborate, and say something like "model YY has high soil moisture and cool temperatures, falling int other VACd category of land-controlled land-atm coupling. This results in <something about surface fluxes> and <something about why this model setup generates VACd vs VACa coupling, or no coupling>"

**Minor:**

- The authors should make it clear throughout the paper how many LSMs are being used. I would suggest saying 4 LSMs the authors distinguish between NOAH & NOAH-MP but sometimes lump NOAH-MP and NOAH-MP-DV together (and sometimes evaluate them separately). It would be more clear if, through the whole paper, they refer to using 4 configurations of LSM: NOAH, NOAH-MP, NOAH-MP-DV, and CLM4; the inclusion of dynamic vegetation in NOAH-MP-DV is pretty fundamentally different than how the other land models distribute vegetation, therefore sometimes lumping it in with NOAH-MP just gets confusing.
- The domain appears to include ocean. If the domain isn't square & doesn't include ocean, please clarify that. If the domain does include ocean (which I assume to be the case), please clarify what method was used for SSTs (prescribed from climatology/satellite observations/reanalysis? Computed?), and how that method may influence the authors' results.
- Line 46: it would be useful if at or before this point, the authors gave a few sentences defining and giving examples of land atmosphere interactions.
- Line 102: "4 different plant functional types" which 4? Regular CLM4 defaults to 14-16 PFTs.
- Line 119: missing from this section how sea surface temperatures are handled
- Line 120-123 (also mentioned above): Are the 3 simulations your "ensemble"? Or, for each LSM, is an ensemble of WRF simulations run? If the former, please clarify. If the later, please provided details (# of ensemble members, how they were initialized)
- Line 121-122: Wording. "The rationale for this decrease in resolution is that this set of simulations constitutes an ensemble of WRF sensitivity experiments". The **rational** is the computational resources. You can still get meaningful results because you're doing a sensitivity study to the LSMs, rather than trying to reproduce obs. The result is a set of 4 WRF simulations that you're calling your ensemble.
- Line 135 / equation 1: This was pretty confusing the first time I read it through, and continues to be a hurdle for the reader through the text. I clear walkthrough of the conditions supporting each VAC situation in the text (and the conditions where there is no VAC) would be super helpful here, along with a description/example of the kind of coupling expected from each category.

- Table 2: a description in the text of the extreme statistics used would be hugely useful. A few of them were talked about near the end of the manuscript, which helped, but introducing them (more than just in the table) here in the methods would make the rest of the manuscript make more sense.
- Line 177: "... using daily data from three Evaluation simulations" what are Evaluation simulations? (Can go look at table S1, but it still doesn't tell me what an "Evaluation simulation" really is, it just tells me what models were used as "Evaluation simulations".
- Line 185: "Atmospheric forcing controls surface processes at middle and high latitudes"

   it controls processes more than land does, but it still appears to have <50% control.</li>
   Please clarify. This is also why a walk-through of the 4 VAC terms in the methods would be useful: what happens if you aren't in a VAC category? Then there isn't strong landatm coupling. Make that clear, and make it clear what the % in figures 1-2 are % of all time that each VAC dominates, or % of time when \*any\* VAC dominates that the specific VAC in question dominates?
- Figure 1 (related to above): These don't add to 100, so can you a add a comment to the methods where you describe vac\_a vac\_d on what happens if none of the 4 are true?
- Figure 1: consider using a different significant mask, e.g. putting dots over the nonsignificant portions, or mask-nonsignificant values with a nan, as it the dots obscure the part of the pattern that matters (I can't tell difference between anything except darkest blue or deepest red when it is under a dot, and those are the only values I really should be looking at)
- Line 193-204; Line 205-216: I found these two paragraphs pretty hard to follow. I think it would be easier to follow if the authors included some discussion about \*why\* each model was under atmospheric / land control in various regions / seasons. E.g. is it the evaporation, or the temperature, or both terms? As it is, I just did a lot of "read one sentence; look at figure 1 (or 2); read next sentence" without being quite sure what was interesting/important about the patterns.

For example:

- Line 208: "episodes over the Mexican coast is higher in CLM4... than ... NOAH-MP-DV simulations in DJF because YYYY"
- Line 212: "the VACc (ie low SM and high TAS, land control due to soil moisture limitation)" or something like that help the reader understand what they're seeing and why what they're seeing is cool!
- Generally, when the authors make a statement about what VACx did, it would be helpful to accompany it with something about what that means for TAS, LH, soil moisture, land control, atm control, etc - give more help to the reader, otherwise the meaning is just lost in a bunch of acronyms, especially for those unfamiliar with VAC metrics where it wouldn't be immediately obvious/intuitive as to what the IMPLICATIONS of being in VACc or VACd are).
- Line 213: at this point I wanted to see a breakdown of VACc and VACd. It is in the supplement maybe point the reader to it here?
- Line 218: "extremes" -> "extreme indices as described in table 2"
- Line 218: "their means" the mean of the extremes? Or just... the means of T and P?
- Line 220-222: "the WRF ensemble mean..." This is confusing since we just went from talking about a bunch of different LSMs in WRF to now discussing a WRF ensemble mean. Are we now talking about the CORDEX? What is the WRF ensemble mean? Do

you mean the mean of all 3 LSMs? Or were ensembles run for each LSM/WRF combination? If the latter, that wasn't clear in the methods.

- Section 4.2 in general: This whole section I was pretty puzzled about what was happening. Are the different LSMs no longer being evaluated/compared? Is this section just laying the ground work for what "normal" WRF looks like, then how it deviates with each LSM will be explored later? If so, please make that clear. If not, what happened to the LSM comparison? I don't think there is anything wrong with the \*content\* of this section, it just needs some additional motivation/transition text to allow the reader to follow why we're no longer hearing about the LSM comparison, which up to this point was the focus.
- Line 230: "Greenland, GRL" Is Greenland actually in your domain? It isn't shown in any of the figures up to this point. And the region highlighted in Figure 4 is not Greenland. Maybe call it "Hudson Bay" instead?
- Figure 3: D = Days? (in color bar legends?) please clarify. Also, consider moving color bar labels below color bars, or putting more horizonal space between plots, so it is clear what unit goes to which color bar
- Figure 4:
  - maybe add x labels to each black-outline-box? hard to go all the way up from the bottom.
  - Are red values in "cold events" very cold, or very not-cold?
  - necessary to add a discussion of each of these extreme metrics to the methods section, more than just the table. Eg "CSDI measures YYYY; a high value means YYYY, while a low value means YYYY", and so on for each metric. (already mentioned this above, but it would be helpful for understanding this figure)
  - See above statement/question re: Greenland
- Line 241: clarify warm events get longer but aren't as hot?
- Line 243: "all simulations represent a similar spatial pattern of the climatology of extreme indices" was this shown? I thought Fig 3 (the climatology) was the average of all the runs. If it wasn't shown, please make it more clear what \*was\* shown, and point to a figure (main text or supplement) to support this sentence.
- Line 249-252: Any insight into why this might be? (CLM4 yields the highest temperatures, NOAH gives the weakest T and P extremes)... do they have super different soil moisture, different surface energy fluxes, produce different boundary layer stabilities...?
- Line 254-256: "simulations show similar spatial patterns…" it would be nice to include some discussion of how much the land model matters (ie are they each behaving similarly, and that is why they look the same?), vs how much the extremes are set by topography, latitude, atmosphere / distance from ocean, etc.
- Line 258: "coldest night in DJF" this is a nice concise description of one of your extreme metrics, nice! It would be great to have something like this for each of the metrics, and have it introduced in the methods (and when you talk about them in the results, rather than just reporting the acronym it'll help the reader understand what is happening and why it is interesting).
- Line 262-264: Again, some discussion of what is causing the spread here would be useful, though maybe beyond the scope of what you'd like to cover in this study. E.g. are the ones that are super hot the ones with low soil moisture?

- Line 283-284: add some discussion... do these places correlate with dry regions? Regions of high topography? What might be generating the spread?
- Line 298-305: I found this section really hard to follow, I think because I wasn't sure what I'm supposed to be taking away from it. Needs more "why" elaboration.
- Figure 5: would be helpful to revist the extreme indices in the caption (Txx = ..., TNn = ... etc.)
- Line 325: Another nice helpful interpretation of the figure/acronym with "less frequent cold nights (TX10p)" thanks! Working more text like that in would help the reader follow!
- Line 328: re: CORDEX simulations Was WRF in CORDEX? Were other regional atm models using CLM? How does your CLM run compare to CORDEX CLM runs? How does your WRF compare to CORDEX WRF?
- Lines 338-341: another place where it would be helpful to do more hand-holding for the reader on why what is being reported here matters
- Line 345-346: precip extremes are more uncertain across CORDEX simulations than WRF simulations -> this would be expected, would it not, as the CORDEX simulations use a variety of different atmospheric models? How does the uncertainty from the choice of atmospheric model compare to the uncertainty from the choice of land model?
- Line 347-348: "...regions with large uncertainties in the simulation of precipitation extremes among the WRF simulations are also identified as areas with large uncertainty across the CORDESX ensemble." This is interesting! Suggests there may be a robust signal.
- Line 354: "results to other model ensembles" ... this seems more like a single atmospheric model study exploring the sensitivity of WRF to perturbed surface fluxes (where the surface fluxes are perturbed by using different LSM components).
- Line 368-369: "The similar uncertainties of extreme evens in the CORDEX ensemble relative to the WRF simulations suggest that the LSM component may be an important source of uncertainty in the CORDEX ensemble." I don't follow the reasoning here. The CORDEX runs use different LSMs yet show similar uncertainties in extremes, so wouldn't that suggest that the LSMs aren't the driver? Please clarify/elaborate.
- Line 370: as with the previous sentence, I'm confused if the authors are trying to talk about how the CORDEX & WRF simulations are similar, or how they're different.
- Line 373: So, the spread in uncertainty within WRF (but with different land models) is bigger than the spread in uncertainty within the CORDEX simulations? Or is the spread within the two sets of simulations being compared to the spread between the two sets of simulations? Please clarify/elaborate.
- Line 384: similarities between WRF and CMIP5 mean elaborate on why they are the same / what is controlling the DJF coldest night / JJA hottest day?
- Line 402: "or in India" -> referring to a specific heat wave (like the 2003 Europe one), or just India in general?
- Line 407: "... depending on the employed LSM component **because YYYY**" (would be nice to have some because/why discussion here)
- Lien 409: What is the authors' recommendation for selecting an LSM? (I don't mean the authors need to pick their favorite, I just would like to see a list of considerations for selecting an appropriate LSM for one's study)
- Line 419: "land atmosphere interactions as measured by YYYY"

- Line 422-423: especially since this is the conclusions, would give a short word sketch on what being in the VACa-d category means.
- Line 430: Include a statement on how much you think your results are the LSM differences, vs 3 instantiations of WRF -> e.g. if you initialized a slightly perturbed CLM-WRF, how different would you expect it to be from your other CLM-WRF, vs how different the various LSM-WRF simulations are?
- Line 432: This sentence would suggest previous comment is mostly LSM dominated, but some explicit discussion of the topic would be nice.

**Typos/grammar:**

- Line 6: "four simulations performed by the WRF model using three different LSMs from 1980 to 2000" this makes it sound like the LSMs are from 1980 to 2000, but I believe the authors mean the simulations are run from 1980-2000, using three different LSMs
- Line 47: "off-line" is "offline" everywhere else
- Line 83: authors "define" NOAH-MP-DV in brackets twice, just need it once
- Line 88-89: I think the second "as" is a typo, but I'm not sure what the authors are trying to say so I don't know how to suggest fixing it. "The NOAH LSM has been extensively used for reanalysis products, as well as for RCM simulations **as** those participating in the CORDEX project..."
- Line 92: missing citation. Perhaps the NOAH technical description? https://ral.ucar.edu/sites/default/files/public/product-tool/unified-noah-lsm/Noah\_LSM\_USERGUIDE\_2.7.1.pdf
- Line 121: "indeed" is a typo. "... counter-intuitive for a RCM experiment; indeed. The ..."
- Line 142-143: typo, I'm not sure what the authors are trying to say. "... clouds and precipitation, which leads to **low vegetation activity likely rising soil moisture**."
- Line 149: "series" -> "time series" (or if that isn't what the authors mean, what is a LH series?)
- Line 162: "techniques techniques" typo
- Line 162-163: "... for the study of future climate trends and climate variability, since **they** have been proven to modify the spatiotemporal consistence of climate models as well as internal feedback mechanisms and conservation terms." This sentence is confusing; in particular, is "they" referring to future climate, or bias removal?
- Line 234: "more frequent cold events than the rest of LSM components" -> "rest of **the** LSM components"
- Line 281: "simulations in about 35 days per year" -> "simulations by about"?
- Line 314: "range among WRF simulations" -> "range among our 4 WRF simulations" (unless you just used 3 confused if NOAH-MP-DV gets used all the time or not)
- Line 323: "WRF ensemble" see earlier comment re: confusion about what your ensemble is
- Line 366: "Thus we compare **each** model's uncertainty..." (insert "each")
- Line 367-268: "despite they used" -> typo. Maybe "despite the fact that they used" ?
- Line 404: "point out to a future" -> "point to a future" (drop "out")

- Line 405-406: typo somewhere, but I'm not sure what the authors are going for thus not sure how to fix it. "Climate model simulations are our best source of information to inform measure against climate change impacts."
- Line 419: "WRF simulations over North America" (specify region is North America)

---

## Referee Comment (RC2) · Anonymous Referee #2 · 4 Jun 2020

Review of "Land Surface Model influence on the simulated climatologies of temperature and precipitation extremes in the WRF v.3.9 model over North America. By Garcia-Garcia et al. Submitted to GMD. Reviewed in June 2020.

This paper is focused on quantifying the uncertainty in the simulation of temperature and precipitation extremes that is associated with the choice of land-surface model (LSM) used in regional climate model (RCM) simulations. The authors performed 4, 34-year climate simulations using WRF driven with NARR boundary conditions. The only difference between each climate simulation was the choice of LSM (NOAH, NOAH-MP,

CLM4, NOAH-MP-VG). They use a single land-atmosphere coupling metric to highlight regional differences in the way the land surface interacts with the atmosphere. They then calculate 16 different temperature and precipitation climate extremes to examine the role of the LSM. Finally they make an attempt to place their work in the context of other model ensembles by comparing climate extremes in their WRF ensemble with some NA-CORDEX models.

This paper is very well written, making it easy to follow. I also appreciate the quality of their figures and color tables. However, as this paper was submitted to a model development journal, I do not believe they include enough discussion of why differences in the LSMs result in differences in land-atmosphere coupling and climate extremes. I suggest this paper be accepted with major revisions.

Major Comment:

1. More information and commentary/insights need to be provided regarding why the different LSM result in variations in land-atmosphere coupling and the VAC index. This could include maps of land cover type/fraction, how surface fluxes are calculated, how soil temperatures are calculated etc. The seasonal cycle of snow cover which will play a role in seasonal transitions to different regimes. Your study shows that the LSM does make a difference, but you need to do more to explain why the models are different (even if it is just hypotheses). This is especially true as you submitted this paper to a Model Development journal – and for this to be useful readers will want to know more about how the LSMs differ and how this could result in changes. Some of the details about this could be in supplemental, but a deeper discussion needs to be included in the paper itself as well.

2. You do not sufficiently link differences in the simulation of land-atmosphere coupling are related to differences in temperature and precipitation extremes. In section 4.3 you do a small amount of work highlighting regions where the VAC index differs and differences occur in the extreme values – but there is no discussion of why/how land

atmosphere coupling may affect the simulation of extremes. This could be included in the introduction, but also in more detail and specific to the LSMs used in this study in section 4.3.

3. The motivation for including NA-CORDEX in this study is not sufficiently clear, and I'm not sure it adds value to the paper. I surmise from section 5.1 that you are trying to show or estimate how much of the uncertainty in temperature and precipitation extremes in multi-model ensembles may be associated with choice of LSM – but as you state there are so many differences in the NA-CORDEX simulations that it's impossible to say what role the LSM actually plays. You make the statement in a few places that the NA-CORDEX models have similar regions with large uncertainties in extremes – but I see more differences between the different model ensembles than similarities.

General Comments.

Need to define what they mean "early on". This paper only focuses on monthly timescales – so that limits the types of extremes that can be studied. All readers will come to this paper with a different assumption of what "extremes" mean. These are outlined in table 2 – but I think saying someplace you are looking at essentially annual maximum values calculated on the daily timestep. Even just a for example inclusion when you mention the climate indices used in the IPCC.

Line 22: The word "interpretation" is not appropriate in this context (here it would mean "explanation" but models don't explain the climate they represent or simulate the climate. I would say "simulation" or "representation".

Line 28: add "the" before IPCC.

Line 31: instead of "affect and are affected by" you could use "are coupled to" and be more clear. Also no comma needed after phenomena.

Paragraph on lines 53-67: At the moment this reads as a "non-sequitur" in the introduction the discussion of LSMs in reanalysis products needs to be linked to the work done

in this work (which does not include analysis of reanalysis products). One option would be to include an explicit statement for why this should be discussed in the introduction. Something along the lines of "examination of the variations in land-atmosphere coupling based on the choice and complexity of the LSM will have implications for weather forecasting and the production of reanalysis products". (or whatever reason you include this information here, if my assumption was incorrect).

Line 68: I suggest adding "coupling & feedbacks" – not all coupling leads to feedbacks per-say.

Lines 113-115: Please provide a justification for why a single year of spin up was used. Is this sufficient for deep soil moisture to spin up? Did you do any testing to see if soil moisture etc. was spun up after one year? What level of soil moisture is important for your study and is that actually spun-up in this time frame?

Line 130-134: This relates to my previous comment that a discussion of what type (e.g. temporal scale) of extreme events this paper is focusing on. You have chosen a LA coupling metric that works on monthly time scales. What type of variability and coupling will you capture using monthly data. Presumably you can calculate the VAC regimes using daily data rather than monthly data – which might include some shorter frequency variations that are lost in the monthly data. Why use a monthly metric when all of your extremes are based on daily maximums/percentiles etc. I'm not saying this was an incorrect choice, it just needs to be explained.

Page 5 – the equations for VAC. I suggest adding "Atmo. Control Coupling" or "Atmo. Control interactions" or something like that – the use of the word control was a little confusing as it could also relate to a "control run".

Line 140: "transitional areas" is not clear. Is this a transition from one regime to another? Why is it a transitional area rather than just a "moisture" limited region where soil moisture plays a larger role. This is the language used in coupling papers such as (Dirmeyer, 2011) or Koster et al, 2009).
[Figure]

Line 143: While the jargon "vegetation activity" may be used with the VAC coupling index – it is a term that is not commonly used, and no meaning to me when reading the paper. Please define what you mean by "vegetation activity" before using the jargon.

Lines 140-145: There is a lot of uncommon jargon (see points above) and I think this section should be revised to make sure people less familiar with using the VAC to estimate coupling can follow what the different regimes are and why they are that way.

Line 160-170: I think this section is critical to include in your paper. Many people who study climate extremes and climate impacts will ask why not just "bias correct" the data. You list very good reasons for this – and I agree you need to look at the absolute value of these terms to really see the differences the LSMs are causing. However – the way this is worded is confusing. I would suggest removing the concept of "bias correction" from this paragraph (lines 161-64) and just discuss the reasons to use absolute and statistical percentile data. Then following that discussion, add in that bias correction is often employed bout would break physical relationships etc. Just a suggestion to improve readability and flow.

Line 175: While I think it is great you include results from NA-CORDEX – I think the context of why you do this analysis needs to be better justified early on (e.g. motivate in the intro better and then remind the reader why you are doing this in the methods).

Line 176: When using existing model simulations, you need to check their data use policy and make sure you appropriately cite the data. There is a DOI that must be included in your paper for NA-COREX (see: https://na-cordex.org/) Mearns, L.O., et al., 2017: The NA-CORDEX dataset, version 1.0. NCAR Climate Data Gateway, Boulder CO, accessed [date], https://doi.org/10.5065/D6SJ1JCH

Paragraph starting on line 175: You have not included enough information about the NA-CORDEX simulations used in your study for the reader to understand the results shown in the paper. Please include the LSMs used and some information about their differences (https://na-cordex.org/rcm-characteristics.html). For example WRF does

use the NOAH model – how different is this from your WRF runs. Some of the models (WRF) use nudging and the others don't, this could cause differences. Also there appears to be more 50km NA-CORDEX simulations with ERA-I boundary conditions (https://na-cordex.org/simulation-matrix.html), why have you only chosen these three?

Section title for 4.1 – You do not do an "evaluation" of the WRF simulations as there are no observations to evaluate the quality of the of the WRF coupling – I think a better word would be "examination" or "comparison"

Figures 1+2: Are all possible cases captured in the 4 VAC categories? Should the sum of all VAC categories equal 100%? This would be useful.

Figure 3: This may be a draft quality issue but it is difficult to read the numbers under the labelbars.

Lines 230-233: This information should be in the figure caption.

Discussion around 345: The WRF NA-CORDEX simulation is a different setup than the model simulations you performed, however it uses the NOAH LSM. Many readers could be curious about how the WRF NA-CORDEX experiment compares with the experiments in this study. Also a discussion of how the NA-CORDEX WRF simulation is different from your WRF simulations would be useful.

---

## Author Comment (AC1) · 14 Jul 2020

Response to Reviewers Document for GMD-2020-86 by Almudena García-García, Francisco José Cuesta-Valero, Hugo Beltrami, Fidel González-Rouco, Elena García-Bustamante and Joel Finnis

We are extremely grateful for the thoughtful and constructive feedback of both reviewers. We really appreciate the quality of the revision, it has improved our new version of the manuscript.

[Figure]

This Response to the Reviewers document provides a complete description of the changes that have been made in response to each individual reviewer comment. Reviewer comments are shown in plain text. Author responses are shown in blue text. All line numbers in the author responses refer to locations in the revised manuscript with changes marked.

Referee 1

Review for gmd-2020-86: "Land Surface Model influence on the simulated climatologies of temperature and precipitation extremes in the WRF v.3.9 model over North America" by García-García et al.

In their article "Land Surface Model influence on the simulated climatologies of temperature and precipitation extremes in the WRF v.3.9 model over North America", García-García et al. use 4 different land surface model (LSM) configurations coupled to the WRF regional atmospheric model over a regional North America domain, to explore the sensitivity of precipitation and temperature extremes to the choice of land model. They find similar overall patterns in the strength of land-atmosphere coupling across their simulations but show that the strength of that coupling can differ significantly across LSMs.

This study is a clear demonstration of how the uncertainty in temperature and precipitation is highly dependent on the choice of land model – a subject too-often overlooked in the study of extremes. The authors provide a valuable contribution to the role of the choice of LSM in the analysis of land-atmosphere coupling and modeling extreme events. Prior to publication, this study would benefit from elaboration on how the authors' results would vary had they explored multiple instantiations of the same LSM-WRF coupling, rather than a single instance of each LSM. Portions of the text were at times dense and confusing – a reworking of these sections (pointed out below) would greatly benefit the reader. Lastly, I would like to see a deeper analysis of the mechanisms driving the differences between the LSM results. The authors may consider this

beyond the scope of their study, but explaining why the LSMs generate different results would be extremely useful to the community, rather than simply pointing out that different LSMs result in different distributions of extremes in temperature and precipitation. Following revision, this manuscript would be appropriate for publication in GMD.

We thank the reviewer for the detail and the quality of this review. We have performed some new calculations and simulations in addition to modify the text in the manuscript to answer the reviewer's point. We think this revision has improved the clarity and quality of our manuscript.

Specific Comments: Major:

- The authors refer to an "ensemble of simulations". To clarify, they did not actually run an ensemble for each model setup, correct? Rather, there are 4 simulations in total evaluated: NOAH/WRF, NOAH-MP/WRF, NOAH-MP-DV/WRF, and CLM/WRF? The authors should clarify the text on this point.

We agree with the referee that the use of the "ensemble" term on the text can be confusing. We have provided more details about what we mean by ensemble (lines 90, 149, 273-274, 280, 311, 404, 414, and caption of Figures 3 and 5).

- Related to (and more important than) the previous comment, the authors do not discuss what variability within a single LSM-WRF framework is expected. Had the authors instead ran an ensemble of, say, 10 NOAH/WRF simulations with slightly different initial conditions, how large a spread in the strength of land-atmosphere coupling and the statistics associated with extreme events would they see? Is the spread we're seeing across the 3/4 LSMs simply due to the fact that WRF was run 4 times, or is it truly a physical response to the physics and mechanisms of the particular LSMs?

To reduce the effect of initial conditions on our results, we used the first year of the simulation as spin-up, which is the spin-up period usually employed in WRF climate simulations to reach the equilibrium of air and soil variables (e.g. Wang and Kotamarthi, 2015, Katragkou et al., 2015 and Barlage et al., 2015). We have included this justification on the text (136-140). We also performed an additional simulation with the CLM4 LSM starting on June 1st, 1979 to test the effect of different initial conditions on our results. The comparison of the WRF solution of monthly latent heat flux and surface air temperature from 1980 to 1981 show very small differences between both simulations (Figures 1 and 2 in this document). Thus, the effect of initial conditions on our results is small.

The sensitivity to initial conditions may also affect the differences in the representation of extreme events between our four simulations, although all of them were initialized on January 1st, 1979 and the first year of the all simulations was discarded. The referee proposes an interesting approach to evaluate the impact of different initial conditions on our results, however performing 10 additional simulations over North America for climatological studies is computationally too expensive. For this reason, we discuss the possible effect of the initial conditions on our results using the available literature (lines 459-465).

- A discussion of why the land-atmosphere coupling strength varies between their simulations would not only help show the differences between the 4 LSM/WRF simulations are actually the result of structural differences in the land model, but would also be greatly beneficial to the reader for understanding why they should care about the spread in LSMs. Including a discussion on this topic would also help the reader select an LSM that appropriately represents the aspect of land-atmosphere coupling they may be interested in studying. I don't mean that the authors need to completely restructure the paper to address this topic. Rather, they often make statements like (Line 203, just an example) "LSM differences in the representation of VACa and VACb probabilities suggest the LSM influence on the evolution of atmospheric conditions". Rather than simply reporting differences in VACa-d, it would be useful for the authors to elaborate, and say something like "model YY has high soil moisture and cool temperatures, falling int other VACd category of land-controlled land-atm coupling. This results in <some-

thing about surface fluxes> and <something about why this model setup generates VACd vs VACa coupling, or no coupling>"

We have modified Section 4.1 addressing the reviewer points. We now comment on the LSM differences in the simulation of extreme latent heat flux and SAT that lead to the differences in the VAC metric (lines 224-269). Additionally, we have related our results of the uncertainty in extreme indices within our four simulations to the land cover and the LSM differences in the representation of vegetation and snow cover (lines 370-403).

Minor:

- The authors should make it clear throughout the paper how many LSMs are being used. I would suggest saying 4 LSMs – the authors distinguish between NOAH NOAH-MP but sometimes lump NOAH-MP and NOAH-MP-DV together (and sometimes evaluate them separately). It would be more clear if, through the whole paper, they refer to using 4 configurations of LSM: NOAH, NOAH-MP, NOAH-MP-DV, and CLM4; the inclusion of dynamic vegetation in NOAH-MP-DV is pretty fundamentally different than how the other land models distribute vegetation, therefore sometimes lumping it in with NOAH- MP just gets confusing.

We have gone though the manuscript addressing this point, stating that we are using three LSMs and we perform four simulations with different LSM configurations (e.g. lines 6, 88, 90 and 149).

- The domain appears to include ocean. If the domain isn't square doesn't include ocean, please clarify that. If the domain does include ocean (which I assume to be the case), please clarify what method was used for SSTs (prescribed from climatology/satellite observations/reanalysis? Computed?), and how that method may influence the authors' results.

SST was prescribed using the NARR product. This means that our simulations is

not ocean/atmospheric coupled, however since we are interested on the influence of the LSM component on the land-atmospheric coupling, and on the representation of extremes, we do not think the use of prescribed SST is affecting our results. We have added a line in the methods indicating this characteristic of the simulations (line 133).

- Line 46: it would be useful if at or before this point, the authors gave a few sentences defining and giving examples of land atmosphere interactions.

Lines 35-52 provides some examples of land-atmosphere interactions and how they affect surface conditions.

- Line 102: "4 different plant functional types" – which 4? Regular CLM4 defaults to 14-16 PFTs.

Thanks for catching that, we have modified this line.

- Line 119: missing from this section – how sea surface temperatures are handled

We have included this information in line 133.

- Line 120-123 (also mentioned above): Are the 3 simulations your "ensemble"? Or, for each LSM, is an ensemble of WRF simulations run? If the former, please clarify. If the later, please provided details ( of ensemble members, how they were initialized)

We performed four simulations with different LSM configurations. We have clarified this point in several lines of the new manuscript (lines 90, 149, 273-274, 280, 311, 404, 414, and caption of Figures 3 and 5).

- Line 121-122: Wording. "The rationale for this decrease in resolution is that this set of simulations constitutes an ensemble of WRF sensitivity experiments". The rational is the computational resources. You can still get meaningful results because you're doing a sensitivity study to the LSMs, rather than trying to reproduce obs. The result is a set of 4 WRF simulations that you're calling your ensemble.

We have clarified this point in the manuscript (lines 145-151).

- Line 135 / equation 1: This was pretty confusing the first time I read it through, and continues to be a hurdle for the reader through the text. I clear walk through of the conditions supporting each VAC situation in the text (and the conditions where there is no VAC) would be super helpful here, along with a description/example of the kind of coupling expected from each category.

The description of the coupling corresponding with each VAC category is explained in lines 161-174. We now include an extra line explaining the no coupling option.

- Table 2: a description in the text of the extreme statistics used would be hugely useful. A few of them were talked about near the end of the manuscript, which helped, but introducing them (more than just in the table) here in the methods would make the rest of the manuscript make more sense.

We have included more general definitions of the indices employed in this study in the Methods section, keeping the technical definitions in Table 2 (see lines 200-206).

- Line 177: "... using daily data from three Evaluation simulations" – what are Evaluation simulations? (Can go look at table S1, but it still doesn't tell me what an "Evaluation simulation" really is, it just tells me what models were used as "Evaluation simulations".

The Evaluation simulations are the CORDEX experiments run with reanalysis forcing. We now defined that on the text (see line 215).

- Line 185: "Atmospheric forcing controls surface processes at middle and high latitudes" – it controls processes more than land does, but it still appears to have <50

Each VAC category yields information about the processes driving energy and water exchanges between the land and the atmosphere, i.e. land-atmosphere interactions. Thus, the VACa and VACb categories indicate that land-atmosphere interactions are controlled by energy restrictions while the VACc and VACd categories are related to limitations in soil water content. The no-coupling option occurs when extremes of latent heat flux and surface air temperature do not coincide in time.

- Figure 1 (related to above): These don't add to 100, so can you a add a comment to the methods where you describe $vac_a - vac_d$ $on what happens if none of the 4 are true?$

- Figure 1: consider using a different significant mask, e.g. putting dots over the non-significant portions, or mask-nonsignificant values with a nan, as it the dots obscure the part of the pattern that matters (I can't tell difference between anything except darkest blue or deepest red when it is under a dot, and those are the only values I really should be looking at)

Good idea, we have masked the non-significant areas with dots so now the significant areas are brighter (Figure 1 and 2).

- Line 193-204; Line 205-216: I found these two paragraphs pretty hard to follow. I think it would be easier to follow if the authors included some discussion about *why* each model was under atmospheric / land control in various regions / seasons. E.g. is it the evaporation, or the temperature, or both terms? As it is, I just did a lot of "read one sentence; look at figure 1 (or 2); read next sentence" without being quite sure what was interesting/important about the patterns. For example: o Line 208: "episodes over the Mexican coast is higher in CLM4... than ... NOAH-MP-DV simulations in DJF because YYYY" o Line 212: "the VACc (ie low SM and high TAS, land control due to soil moisture limitation)" or something like that – help the reader understand what they're seeing and why what they're seeing is cool! o Generally, when the authors make a statement about what VACx did, it would be helpful to accompany it with something about what that means for TAS, LH, soil moisture, land control, atm control, etc - give more help to the reader, otherwise the meaning is just lost in a bunch of acronyms, especially for those unfamiliar with VAC metrics where it wouldn't be immediately obvious/intuitive as to what the IMPLICATIONS of being in VACc or VACd are).

We have re-written this section addressing the referee comment (lines 235-269).

- Line 213: at this point I wanted to see a breakdown of VACc and VACd. It is in the supplement – maybe point the reader to it here?

In line 214 we referred to Figures S3 and S4 in the supplementary information. We have modified several references to make clear that we mean figures in the supplementary information (e.g. lines 239, 245, 247, 277, 376 and 390).

- Line 218: "extremes" -> "extreme indices as described in table 2"

We have modified this line (line 275).

- Line 218: "their means" – the mean of the extremes? Or just... the means of T and P?

We meant the mean of each extreme index (lines 275-276).

- Line 220-222: "the WRF ensemble mean..." This is confusing since we just went from talking about a bunch of different LSMs in WRF to now discussing a WRF ensemble mean. Are we now talking about the CORDEX? What is the WRF ensemble mean? Do you mean the mean of all 3 LSMs? Or were ensembles run for each LSM/WRF combination? If the latter, that wasn't clear in the methods.

By WRF ensemble mean, we meant the mean of the four simulations performed with the WRF model. We have clarified that in several parts of the new version of the manuscript (lines 90, 149, 273-274, 280, 311, 404, 414, and caption of Figures 3 and 5).

- Section 4.2 in general: This whole section I was pretty puzzled about what was happening. Are the different LSMs no longer being evaluated/compared? Is this section just laying the ground work for what "normal" WRF looks like, then how it deviates with each LSM will be explored later? If so, please make that clear. If not, what happened to the LSM comparison? I don't think there is anything wrong with the *content* of this section, it just needs some additional motivation/transition text to allow the reader to follow why we're no longer hearing about the LSM comparison, which up to this point was the focus.

We have added an additional (first) paragraph in this section to give a little bit of context as the reviewer suggested (lines 271-274). In this section, we continue focused on LSM differences, comparing the extreme indices values in our four WRF simulations. But first, we analyze the spatial features of the climatology of extreme temperature and precipitation indices as simulated by the mean of the four WRF simulations with different LSM configurations (WRF ensemble mean) and by each LSM simulation separately.

- Line 230: "Greenland, GRL" - Is Greenland actually in your domain? It isn't shown in any of the figures up to this point. And the region highlighted in Figure 4 is not Greenland. Maybe call it "Hudson Bay" instead?

Greenland is not included in our analysis. However, we called Greenland to our northeast subdomain to be consistent with the spatial classification carried out by Giorgi and Francisco (2000) on which we based our boundary criteria. We would like to continue using this name for the subdomain.

- Figure 3: D = Days? (in color bar legends?) please clarify. Also, consider moving color bar labels below color bars, or putting more horizonal space between plots, so it is clear what unit goes to which color bar

Thanks for the suggestion. We have modified the figure accordingly (see Figure 3 in the new version of the manuscript).

- Figure 4: o maybe add x labels to each black-outline-box? hard to go all the way up from the bottom.

Because of space reasons, we can not include the x-axis below each panel, it will make the figure extremely long.

o Are red values in "cold events" very cold, or very not-cold?

Red always refers to the hottest and blue always refers to the coldest for cold and warm events, as explained in the caption of Figure 4. Thus, the higher value of TX10p (number of cold days) and the higher value of CSDI (number of consecutive cold nights)
are represented in dark blue for example.

o necessary to add a discussion of each of these extreme metrics to the methods section, more than just the table. Eg "CSDI measures YYYY; a high value means YYYY, while a low value means YYYY", and so on for each metric. (already mentioned this above, but it would be helpful for understanding this figure)

We have included more general definitions of the indices in the methodology according with the previous comment (lines 200-206).

o See above statement/question re: Greenland

We would like to keep the GRD name for the subdomain to be consistent with the classification of Giorgi and Francisco (2000).

- Line 241: clarify – warm events get longer but aren't as hot?

Right. We have clarified that in the text (line 302-303).

- Line 243: "all simulations represent a similar spatial pattern of the climatology of extreme indices" – was this shown? I thought Fig 3 (the climatology) was the average of all the runs. If it wasn't shown, please make it more clear what *was* shown, and point to a figure (main text or supplement) to support this sentence.

It was shown in figures S10, S11 and S12 in the supplementary information. We have included a reference to these figures in the sentence pointed by the reviewer (line 277 and 312).

- Line 249-252: Any insight into why this might be? (CLM4 yields the highest temperatures, NOAH gives the weakest T and P extremes)... do they have super different soil moisture, different surface energy fluxes, produce different boundary layer stabilities...?

We have commented on the LSM differences in the simulation of latent heat flux that lead to the differences in the land-atmosphere metric and therefore to the differences in extreme events through land-atmosphere feedbacks (lines 310-330).

- Line 254-256: "simulations show similar spatial patterns..." – it would be nice to include some discussion of how much the land model matters (ie are they each behaving similarly, and that is why they look the same?), vs how much the extremes are set by topography, latitude, atmosphere / distance from ocean, etc.

This is probably related to the similar spatial pattern of land-atmosphere interactions shown by all LSM simulations and the different degree of coupling at some locations, which likely means that there are other factors beyond the LSM component generating the spatial pattern of extremes and land-atmosphere interactions, such as the topography, land cover and atmosphere parameterizations that are the same for the four simulations. We include a few lines about this in the new version of the manuscript (see lines 310-315).

- Line 258: "coldest night in DJF" – this is a nice concise description of one of your extreme metrics, nice! It would be great to have something like this for each of the metrics, and have it introduced in the methods (and when you talk about them in the results, rather than just reporting the acronym – it'll help the reader understand what is happening and why it is interesting).

We have included more general definitions of the indices in the methodology and in the discussion of the results as suggested (lines 200-206, 346 and 358 and in the caption of Figures 3, 5 and 6).

- Line 262-264: Again, some discussion of what is causing the spread here would be useful, though maybe beyond the scope of what you'd like to cover in this study. E.g. are the ones that are super hot the ones with low soil moisture?

We have investigated deeper into the LSM differences and provided a few hypotheses about that in the text (lines 365-397).

- Line 283-284: add some discussion... do these places correlate with dry regions? Regions of high topography? What might be generating the spread?

We have related these results to the land cover and the differences in the representation of vegetation between LSM configurations (lines 370-403).

- Line 298-305: I found this section really hard to follow, I think because I wasn't sure what I'm supposed to be taking away from it. Needs more "why" elaboration.

We have revised this section, changing most of the text (lines 370-403).

- Figure 5: would be helpful to revist the extreme indices in the caption (Txx = ..., TNn = ... etc.)

We have included a sort definition of the indices in the captions of Figures 3, 5 and 6.

- Line 325: Another nice helpful interpretation of the figure/acronym with "less frequent cold nights (TX10p)" – thanks! Working more text like that in would help the reader follow!

We have gone through the text and included more of these references (346 and 358 and in the caption of Figures 3, 5 and 6).

- Line 328: re: CORDEX simulations - Was WRF in CORDEX? Were other regional atm models using CLM? How does your CLM run compare to CORDEX CLM runs? How does your WRF compare to CORDEX WRF?

Indeed a simulation performed with the WRF model using the NOAH LSM is included in the CORDEX ensemble formed by three simulations. The comparison of our WRF-NOAH simulation and the one from the CORDEX ensemble (second column in Figures S11-S13 and third column in Figures S17-S19 in the supplementary information) show similarities in the spatial pattern of the extreme indices with some differences in the index values, like it is shown for the other two CORDEX simulations. The spatial similarities suggest the influence of topography, land cover and the latitudinal gradient on the spatial features of these results. The specific differences between our WRF-NOAH simulation and that included in the CORDEX project, are likely related to the different boundary conditions and the nudging technique used by the CORDEX WRF-NOAH

simulation to match the employed reanalysis product. We have included a few lines about this in the discussion section (lines 476-484).

- Lines 338-341: another place where it would be helpful to do more hand-holding for the reader on why what is being reported here matters

These lines show examples of areas were the uncertainties in the WRF and CORDEX ensembles are similar. Since our simulations only differ in the LSM configuration we conclude that differences in the LSM components can also be responsible for some of the differences between the CORDEX simulations, and therefore the LSM component can be an important source of uncertainty in inter-model ensembles. We have made this clearer at the end of this paragraph (lines 447-450).

- Line 345-346: precip extremes are more uncertain across CORDEX simulations than WRF simulations -> this would be expected, would it not, as the CORDEX simulations use a variety of different atmospheric models? How does the uncertainty from the choice of atmospheric model compare to the uncertainty from the choice of land model?

Right, atmospheric parameterizations are expected to play a crucial role in the simulation of precipitation events, and therefore the range of precipitation indices among the CORDEX simulations is expected to be larger than the range among our WRF simulations. We have modified the text to include this point explicitly (lines 445-447). However, the comparison between the effect of atmospheric parameterizations and the effect of the land surface model on the representation of extreme events requires to perform another set of simulations with different atmospheric parameterizations, which is computationally too expensive.

- Line 347-348: "...regions with large uncertainties in the simulation of precipitation extremes among the WRF simulations are also identified as areas with large uncertainty across the CORDESX ensemble." This is interesting! Suggests there may be a robust signal.

Thanks, we have tried to emphasize that on the text (lines 466-475).

- Line 354: "results to other model ensembles" ... this seems more like a single at-mospheric model study exploring the sensitivity of WRF to perturbed surface fluxes (where the surface fluxes are perturbed by using different LSM components).

Right, we have modified this sentence to avoid possible confusion (lines 451).

- Line 368-369: "The similar uncertainties of extreme evens in the CORDEX ensemble relative to the WRF simulations suggest that the LSM component may be an important source of uncertainty in the CORDEX ensemble." I don't follow the reasoning here. The CORDEX runs use different LSMs yet show similar uncertainties in extremes, so wouldn't that suggest that the LSMs aren't the driver? Please clarify/elaborate.

The agreement in the placement of areas with large uncertainties in the representation of extreme events within the CORDEX ensemble and those within our WRF simula-tions suggests that the uncertainties in these areas may arise from similar causes. Our simulations only differ in the configuration of the LSM component. Therefore, the dif-ferences between LSM configurations should be contributing to the uncertainties in the representation of land-atmosphere interactions and extreme events within both, our WRF simulations and the CORDEX ensemble. We have included these lines in the new version of the manuscript (lines 466-475).

- Line 370: as with the previous sentence, I'm confused if the authors are trying to talk about how the CORDEX WRF simulations are similar, or how they're different.

We have modified the end of this paragraph to clarify this point (lines 466-475).

- Line 373: So, the spread in uncertainty within WRF (but with different land models) is bigger than the spread in uncertainty within the CORDEX simulations? Or is the spread within the two sets of simulations being compared to the spread between the two sets of simulations? Please clarify/elaborate.

We are comparing the uncertainty within the WRF simulations with the uncertainty

within the CORDEX ensemble. That is, the difference between the range among the WRF simulations and the range among the CORDEX simulations. We have clarified this in the text (485-489).

- Line 384: similarities between WRF and CMIP5 mean – elaborate on why they are the same / what is controlling the DJF coldest night / JJA hottest day?

As commented above, the similarities in the spatial pattern of extreme indices between our WRF simulations, the CORDEX ensemble and the CMIP5 ensemble indicate that the topography, land cover and latitudinal gradient are driving these spatial features. We have elaborated on this in the new version of the manuscript (lines 500-502).

- Line 402: "or in India" -> referring to a specific heat wave (like the 2003 Europe one), or just India in general?

We now provide with a more specific example (line 519).

- Line 407: "... depending on the employed LSM component because YYYY" (would be nice to have some because/why discussion here)

Added (lines 525).

- Lien 409: What is the authors' recommendation for selecting an LSM? (I don't mean the authors need to pick their favorite, I just would like to see a list of considerations for selecting an appropriate LSM for one's study)

Since we are not comparing with observations of land-atmosphere interactions or extreme events, we can not select the LSM that best works for North America. That will constitute the follow up of this study. Nonetheless, what we can state the importance of selecting the LSM configuration in model simulations because it may strongly affect the results of the experiment. Additionally, studies based on multi-model ensembles and reanalyses should include a variety of LSM configurations to account for the associated uncertainty or to test the performance of the selected LSM component before performing the whole simulation. We have included a little bit of this discussion in the

new version of the manuscript (lines 525-531).

- Line 419: "land atmosphere interactions as measured by YYYY"

Included (line 541).

- Line 422-423: especially since this is the conclusions, would give a short word sketch on what being in the VACa-d category means.

We now provide with more information about the VAC index in the Conclusions section, focused on the VACa-b and VACc-d classification (lines 541-544).

- Line 430: Include a statement on how much you think your results are the LSM differences, vs 3 instantiations of WRF -> e.g. if you initialized a slightly perturbed CLM-WRF, how different would you expect it to be from your other CLM-WRF, vs how different the various LSM-WRF simulations are?

We have included a few lines in the Discussion about the possible effect of the initial conditions on our results based on the literature (lines 459-465).

- Line 432: This sentence would suggest previous comment is mostly LSM dominated, but some explicit discussion of the topic would be nice.

We have included a few sentences about that in the new version of the manuscript (lines 555-559).

Typos/grammar:

- Line 6: "four simulations performed by the WRF model using three different LSMs from 1980 to 2000" – this makes it sound like the LSMs are from 1980 to 2000, but I believe the authors mean the simulations are run from 1980-2000, using three different LSMs

We have modified that sentence (line 6).

- Line 47: "off-line" is "offline" everywhere else

We have corrected that typo (line 59).

- Line 83: authors "define" NOAH-MP-DV in brackets twice, just need it once

Right. We have removed the second one (line 103).

- Line 88-89: I think the second "as" is a typo, but I'm not sure what the authors are trying to say so I don't know how to suggest fixing it. "The NOAH LSM has been extensively used for reanalysis prodcuts, as well as for RCM simulations as those participating in the CORDEX project..."

We have replaced "as" by "such as" to make it clearer (line 108).

- Line 92: missing citation. Perhaps the NOAH technical description? https://ral.ucar.edu/sites/default/files/public/product-tool/unified-noah-lsm/Noah$_{LSM_USERGUIDE_2}$.7.1.$pdf$

Right, we now include this reference (line 112).

- Line 121: "indeed" is a typo. "... counter-intuitive for a RCM experiment; indeed. The ..."

Thanks for catching that, we have corrected the typo (line 146).

- Line 142-143: typo, I'm not sure what the authors are trying to say. "... clouds and precipitation, which leads to low vegetation activity likely rising soil moisture."

We have modified this sentence to make it clearer (line 168).

- Line 149: "series" -> "time series" (or if that isn't what the authors mean, what is a LH series?)

We meant "time series". We have corrected that in line 176.

- Line 162: "techniques techniques" typo

We have corrected this typo (line 193).

- Line 162-163: "... for the study of future climate trends and climate variability, since they have been proven to modify the spatiotemporal consistence of climate models as well as internal feedback mechanisms and conservation terms." This sentence is confusing; in particular, is "they" referring to future climate, or bias removal?

Agreed and changed (line 194).

- Line 234: "more frequent cold events than the rest of LSM components" -> "rest of the LSM components"

Corrected (line 294).

- Line 281: "simulations in about 35 days per year" -> "simulations by about" ?

Changed (line 360).

- Line 314: "range among WRF simulations" -> "range among our 4 WRF simulations" (unless you just used 3 – confused if NOAH-MP-DV gets used all the time or not)

Yes, we always use the four WRF simulations including the NOAH-MP-DV. We have modified the indicated sentence as suggested (line 404).

- Line 323: "WRF ensemble" – see earlier comment re: confusion about what your ensemble is

Changed (lines 414).

- Line 366: "Thus we compare each model's uncertainty..." (insert "each")

We have modified the sentence to clarify what we meant (line 468).

- Line 367-268: "despite they used" -> typo. Maybe "despite the fact that they used" ?

Changed (line 470).

- Line 404: "point out to a future" -> "point to a future" (drop "out")

Thanks for catching this typo. Corrected (line 521).

- Line 405-406: typo somewhere, but I'm not sure what the authors are going for thus not sure how to fix it. "Climate model simulations are our best source of information to inform measure against climate change impacts."

Changed (line 523).

- Line 419: "WRF simulations over North America" (specify region is North America)

Included (line 540).

REFERENCES

Barlage, M., Tewari, M., Chen, F. et al. The effect of groundwater interaction in North American regional climate simulations with WRF/Noah-MP. Climatic Change 129, 485–498 (2015). https://doi.org/10.1007/s10584-014-1308-8

Katragkou, E., García Díez, M., Vautard, R., Sobolowski, S. P., Zanis, P., Alexandri, G. Et al. (2015). Regional climate hindcast simulations within EURO-CORDEX: evaluation of a WRF multi-physics ensemble. Geosci. Model Dev., 8,:603-618. doi:10.5194/gmd-8-603-2015

Seneviratne, S. I., Corti, T., Davin, E. L., Hirschi, M., Jaeger, E. B., Lehner, I., ... Teuling, A. J. (2010). Investigating soil moisture–climate interactions in a changing climate: A review. Earth-Science Reviews, 99(3-4), 125-161.

Wang, J. and Kotamarthi, V.R. (2015), High-resolution dynamically downscaled projections of precipitation in the mid and late 21st century over North America. Earth's Future, 3: 268-288. doi:10.1002/2015EF000304

[Figure]

[Figure]

**Fig. 1.** Monthly time series of latent heat flux averaged over North America (NA) and the subdo-
mains included in the manuscript. The black line corresponds with the WRF-CLM4 simulation
employed for our analysi

**Fig. 2.** Monthly time series of surface air temperature averaged over North America (NA) and the subdomains included in the manuscript. The black line corresponds with the WRF-CLM4 simulation employed for our

---

## Author Comment (AC2) · 14 Jul 2020

Response to Reviewers Document for GMD-2020-86 by Almudena García-García, Francisco José Cuesta-Valero, Hugo Beltrami, Fidel González-Rouco, Elena García-Bustamante and Joel Finnis

We are extremely grateful for the thoughtful and constructive feedback of both reviewers. We really appreciate the quality of the revision, it has improved our new version of the manuscript.

This Response to the Reviewers document provides a complete description of the changes that have been made in response to each individual reviewer comment. Reviewer comments are shown in plain text. Author responses are shown in blue text. All line numbers in the author responses refer to locations in the revised manuscript with changes marked.

Referee 2

Review of "Land Surface Model influence on the simulated climatologies of temperature and precipitation extremes in the WRF v.3.9 model over North America. By Garcia-Garcia et al. Submitted to GMD. Reviewed in June 2020.

This paper is focused on quantifying the uncertainty in the simulation of temperature and precipitation extremes that is associated with the choice of land-surface model (LSM) used in regional climate model (RCM) simulations. The authors performed 4, 34-year climate simulations using WRF driven with NARR boundary conditions. The only difference between each climate simulation was the choice of LSM (NOAH, NOAH-MP, CLM4, NOAH-MP-VG). They use a single land-atmosphere coupling metric to highlight regional differences in the way the land surface interacts with the atmosphere. They then calculate 16 different temperature and precipitation climate extremes to examine the role of the LSM. Finally they make an attempt to place their work in the context of other model ensembles by comparing climate extremes in their WRF ensemble with some NA-CORDEX models.

This paper is very well written, making it easy to follow. I also appreciate the quality of their figures and color tables. However, as this paper was submitted to a model development journal, I do not believe they include enough discussion of why differences in the LSMs result in differences in land-atmosphere coupling and climate extremes. I suggest this paper be accepted with major revisions.

As it was the case for the revision of referee 1, we thank the reviewer for the detail and the quality of this review. Some of these comments were also pointed out by referee 1.

We have addressed these points by performing some new calculations and simulations in addition to modify the text in the manuscript. We think this revision has improved the clarity and quality of our manuscript.

Major Comment:

1. More information and commentary/insights need to be provided regarding why the different LSM result in variations in land-atmosphere coupling and the VAC index. This could include maps of land cover type/fraction, how surface fluxes are calculated, how soil temperatures are calculated etc. The seasonal cycle of snow cover which will play a role in seasonal transitions to different regimes. Your study shows that the LSM does make a difference, but you need to do more to explain why the models are different (even if it is just hypotheses). This is especially true as you submitted this paper to a Model Development journal – and for this to be useful readers will want to know more about how the LSMs differ and how this could result in changes. Some of the details about this could be in supplemental, but a deeper discussion needs to be included in the paper itself as well.

We have modified sections 4.1, 4.2 and 4.3 to provide with a more comprehensive explanation about differences between LSMs in the representation of the VAC index and the climatology of extremes. Additionally, we have plotted the land use categories employed in the four simulations to study the role of the different LSM representations of land cover on the uncertainties in the representation of extreme indices among our four simulations. This comparison allowed us to identify some coincidences between vegetation and snow cover and areas with large uncertainty in the simulations of extremes. Thus, LSM differences in the representation of vegetation and snow cover are likely affecting the simulation of land-atmosphere interactions and consequently the simulation of extremes.

2. You do not sufficiently link differences in the simulation of land-atmosphere coupling are related to differences in temperature and precipitation extremes. In section 4.3

you do a small amount of work highlighting regions where the VAC index differs and differences occur in the extreme values – but there is no discussion of why/how land atmosphere coupling may affect the simulation of extremes. This could be included in the introduction, but also in more detail and specific to the LSMs used in this study in section 4.3.

LSM differences in the representation of land cover and soil conditions will lead to a different representation of energy fluxes at the land surface, affecting atmospheric processes. The different LSM representation of latent heat flux will affect the simulation of surface temperatures in the following way: a decrease in latent heat flux will likely mean an increase in the energy available for sensible heat flux, which is directly related to the air-ground temperature gradient. The increase in sensible heat flux yields an increase in this temperature gradient, likely leading to changes in air temperatures (Seneviratne et al., 2010). Meanwhile changes in latent heat flux originated from the different LSM components yield changes in the atmospheric water content, possibly leading to changes in the formation of clouds and precipitation (Seneviratne et al., 2010). We have included this discussion in the introduction and in section 4.3 in the new version of the manuscript (lines 35-44 and 318-330).

3. The motivation for including NA-CORDEX in this study is not sufficiently clear, and I'm not sure it adds value to the paper. I surmise from section 5.1 that you are trying to show or estimate how much of the uncertainty in temperature and precipitation extremes in multi-model ensembles may be associated with choice of LSM – but as you state there are so many differences in the NA-CORDEX simulations that it's impossible to say what role the LSM actually plays. You make the statement in a few places that the NA-CORDEX models have similar regions with large uncertainties in extremes – but I see more differences between the different model ensembles than similarities.

We still think that the comparison of our simulations with the CORDEX simulations is interesting to present the hypothesis that the LSM component is contributing to the uncertainty in inter-model ensembles. The similarities in the areas with large uncertainty in the simulation of extreme indices within our four WRF simulations and the CORDEX simulations suggest that the LSM can be an important source of uncertainty in inter-model ensembles. Therefore, some caution should be taken when selecting a multi-model ensemble to make sure you include a variety of LSM components so you account for the associated uncertainty. We have tried to emphasize this in the new version of the manuscript (lines 63-66, 89-91, 445-450, 466-475 and 525-531).

General Comments.

Need to define what they mean "early on". This paper only focuses on monthly timescales – so that limits the types of extremes that can be studied. All readers will come to this paper with a different assumption of what "extremes" mean. These are outlined in table 2 – but I think saying someplace you are looking at essentially annual maximum values calculated on the daily timestep. Even just a for example inclusion when you mention the climate indices used in the IPCC.

We have included a sentence to clarify that we use these extreme indices at climatological scales at the end of the first paragraph in the Introduction (lines 34). We have also defined all extreme indices in the methodology using more general words to make sure the reader understands what we are representing (lines 200-206).

Line 22: The word "interpretation" is not appropriate in this context (here it would mean "explanation" but models don't explain the climate they represent or simulate the climate. I would say "simulation" or "representation".

Agreed and changed (line 26).

Line 28: add "the" before IPCC.

Done (line 32).

Line 31: instead of "affect and are affected by" you could use "are coupled to" and be more clear. Also no comma needed after phenomena.

Changed (line 35).

Paragraph on lines 53-67: At the moment this reads as a "non-sequitur" in the introduction the discussion of LSMs in reanalysis products needs to be linked to the work done in this work (which does not include analysis of reanalysis products). One option would be to include an explicit statement for why this should be discussed in the introduction. Something along the lines of "examination of the variations in land-atmosphere coupling based on the choice and complexity of the LSM will have implications for weather forecasting and the production of reanalysis products". (or whatever reason you include this information here, if my assumption was incorrect).

Agreed. We have worked on the connection of this paragraph with the introduction (lines 66-69).

Line 68: I suggest adding "coupling feedbacks" – not all coupling leads to feedbacks per-say.

Agreed and changed (line 83).

Lines 113-115: Please provide a justification for why a single year of spin up was used. Is this sufficient for deep soil moisture to spin up? Did you do any testing to see if soil moisture etc. was spun up after one year? What level of soil moisture is important for your study and is that actually spun-up in this time frame?

We used the one year spin-up because it is the spin up duration used in previous WRF climate simulations to reach the equilibrium of air and soil variables, such as those in Wang and Kotamarthi, (2015), Katragkou et al., (2015) and Barlage et al., (2015). We have included this justification on the text (lines 135-140). To address the reviewer query, we also performed an additional simulation with the CLM4 LSM starting on June 1st 1979. The comparison of the WRF solution of monthly latent heat flux and surface air temperature from 1980 to 1981 show very small differences between both simulations (Figures 1 and 2 in this document). Thus, the effect of initial conditions on

our results is small.

Line 130-134: This relates to my previous comment that a discussion of what type (e.g. temporal scale) of extreme events this paper is focusing on. You have chosen a LA coupling metric that works on monthly time scales. What type of variability and coupling will you capture using monthly data. Presumably you can calculate the VAC regimes using daily data rather than monthly data – which might include some shorter frequency variations that are lost in the monthly data. Why use a monthly metric when all of your extremes are based on daily maximums/percentiles etc. I'm not saying this was an incorrect choice, it just needs to be explained.

We used the VAC metric at monthly scales with the 30th and 70th percentiles as in Sippel et al., 2013, where the authors demonstrated that this monthly metric is useful for the analysis of daily extreme events at climatology scales by a statistical analysis and the comparison of the VAC metric with another correlation metric. We have included this justification on the text (lines 162-164). Additionally, since we are interested in the climatology of extreme events estimating the mean of all extreme indices for the analysis period, we do not think the monthly VAC metric is losing relevant information for our study.

Page 5 – the equations for VAC. I suggest adding "Atmo. Control Coupling" or "Atmo. Control interactions" or something like that – the use of the word control was a little confusing as it could also relate to a "control run".

Agreed and changed (Equation 1, page 7).

Line 140: "transitional areas" is not clear. Is this a transition from one regime to another? Why is it a transitional area rather than just a "moisture" limited region where soil moisture plays a larger role. This is the language used in coupling papers such as (Dirmeyer, 2011) or Koster et al, 2009).

Right, the "transition areas" term can be confusing. We have changed this term by

"water limited areas" (line 165).

Line 143: While the jargon "vegetation activity" may be used with the VAC coupling index – it is a term that is not commonly used, and no meaning to me when reading the paper. Please define what you mean by "vegetation activity" before using the jargon.

We have replaced all vegetation activity terms by vegetation photosynthetic activity to be more accurate (lines 168, 170 and 173).

Lines 140-145: There is a lot of uncommon jargon (see points above) and I think this section should be revised to make sure people less familiar with using the VAC to estimate coupling can follow what the different regimes are and why they are that way.

We have revised this section, and we think it is clear now (lines 161-174).

Line 160-170: I think this section is critical to include in your paper. Many people who study climate extremes and climate impacts will ask why not just "bias correct" the data. You list very good reasons for this – and I agree you need to look at the absolute value of these terms to really see the differences the LSMs are causing. However – the way this is worded is confusing. I would suggest removing the concept of "bias correction" from this paragraph (lines 161-64) and just discuss the reasons to use absolute and statistical percentile data. Then following that discussion, add in that bias correction is often employed bout would break physical relationships etc. Just a suggestion to improve readability and flow.

Great suggestion, we have changed the flow of the paragraph in the new version of the manuscript (lines 189-200).

Line 175: While I think it is great you include results from NA-CORDEX – I think the context of why you do this analysis needs to be better justified early on (e.g. motivate in the intro better and then remind the reader why you are doing this in the methods).

We have improved the context for the use of the CORDEX simulations in the introduction and in the methodology as suggested (lines 63-66, 89-91, and 211-215).

Line 176: When using existing model simulations, you need to check their data use policy and make sure you appropriately cite the data. There is a DOI that must be included in your paper for NA-CORDEX (see: https://na-cordex.org/) Mearns, L.O., et al., 2017: The NA-CORDEX dataset, version 1.0. NCAR Climate Data Gateway, Boulder CO, accessed [date], https://doi.org/10.5065/D6SJ1JCH

We have included this reference in the main text (see line 214).

Paragraph starting on line 175: You have not included enough information about the NA-CORDEX simulations used in your study for the reader to understand the results shown in the paper. Please include the LSMs used and some information about their differences (https://na-cordex.org/rcm-characteristics.html). For example WRF does use the NOAH model – how different is this from your WRF runs. Some of the models (WRF) use nudging and the others don't, this could cause differences. Also there appears to be more 50km NA-CORDEX simulations with ERA-I boundary conditions (https://na-cordex.org/simulation-matrix.html), why have you only chosen these three?

The NA-CORDEX information about RCM differences was summarized in table S1, now included in the main text as Table 3 in the new version of the manuscript. As the reviewer indicates there is a WRF simulation employing the NOAH LSM. However, this simulation used nudging to match the boundary conditions from the ERA-Interim reanalysis. The comparison of our WRF-NOAH simulation and the one included in the CORDEX ensemble (second column in Figures S11-13 and third column in Figures S17-S19 in the supplementary information) shows similarities in the spatial pattern of the extreme indices with some differences in the index values, like it is shown for the other 2 CORDEX simulations. The spatial similarities suggest that the topography, land cover and the latitudinal gradient may be driving the spatial features of these results. The specific differences between our WRF-NOAH simulation and that included in the CORDEX project are probably driven by the boundary conditions because in addition to be different, the CORDEX WRF simulation used nudging techniques to match the employed reanalysis product. We have included this discussion in the new version of

the manuscript (lines 476-484).

Furthermore, the NA-CORDEX project includes evaluation simulation for three extra RCMs (the RCM4, the HIRHAM5 and the REGCM4 RCMs). However, all these simulations start in 1989 and end in 2009 or 2011. The use of these simulations would reduce our analysis period significantly, so we decided to use the three simulations providing data for the analysis period of our simulations. We have included this justification on the text (lines 218-219).

Section title for 4.1 – You do not do an "evaluation" of the WRF simulations as there are no observations to evaluate the quality of the of the WRF coupling – I think a better word would be "examination" or "comparison"

We have gone through the text and replaced the "evaluation" word by a more appropriate term in the cases where it can be confusing (e.g. lines 5, 26, 31, 85, 93, 147 and 223).

Figures 1+2: Are all possible cases captured in the 4 VAC categories? Should the sum of all VAC categories equal 100

Figure 3: This may be a draft quality issue but it is difficult to read the numbers under the labelbars.

We have modified this Figure to enlarge the color scales and improve the quality of the figure.

Lines 230-233: This information should be in the figure caption.

We have included these lines in the caption of the figure.

Discussion around 345: The WRF NA-CORDEX simulation is a different setup than the model simulations you performed, however it uses the NOAH LSM. Many readers could be curious about how the WRF NA-CORDEX experiment compares with the experiments in this study. Also a discussion of how the NA-CORDEX WRF simulation

is different from your WRF simulations would be useful.

The comparison of these simulations (second column in Figures S11-12 and third column in Figures S17-S19 in the supplementary information) show similarities in the spatial pattern of the extremes with some differences in the index values, like it is shown for the other 2 CORDEX simulations. The spatial similarities suggest that the topography, land cover and the latitudinal gradient may be driving the spatial features of these results. The specific differences between our WRF-NOAH simulation and that included in the CORDEX project are probably related to the different boundary conditions because the CORDEX WRF simulation uses nudging techniques to match the employed reanalysis product. We have included this discussion in the new version of the manuscript (lines 476-484).

REFERENCES

Barlage, M., Tewari, M., Chen, F. et al. The effect of groundwater interaction in North American regional climate simulations with WRF/Noah-MP. Climatic Change 129, 485–498 (2015). https://doi.org/10.1007/s10584-014-1308-8

Katragkou, E., García Díez, M., Vautard, R., Sobolowski, S. P., Zanis, P., Alexandri, G. Et al. (2015). Regional climate hindcast simulations within EURO-CORDEX: evaluation of a WRF multi-physics ensemble. Geosci. Model Dev., 8,:603-618. doi:10.5194/gmd-8-603-2015

Seneviratne, S. I., Corti, T., Davin, E. L., Hirschi, M., Jaeger, E. B., Lehner, I., ... Teuling, A. J. (2010). Investigating soil moisture–climate interactions in a changing climate: A review. Earth-Science Reviews, 99(3-4), 125-161.

Wang, J. and Kotamarthi, V.R. (2015), High-resolution dynamically downscaled projections of precipitation in the mid and late 21st century over North America. Earth's Future, 3: 268-288. doi:10.1002/2015EF000304

[Figure]

**Fig. 1.** Monthly time series of latent heat flux averaged over North America (NA) and the subdomains included in the manuscript. The black line corresponds with the WRF-CLM4 simulation employed for our analysi

[Figure]

**Fig. 2.** Monthly time series of surface air temperature averaged over North America (NA) and the subdomains included in the manuscript. The black line corresponds with the WRF-CLM4 simulation employed for our

---

## Author Response (AR2)

**Response to Reviewers Document for GMD-2020-86 by Almudena García-García, Francisco José Cuesta-Valero, Hugo Beltrami, Fidel González-Rouco, Elena García-Bustamante and Joel Finnis**

**We are grateful for the thoughtful and constructive feedback of both reviewers.**

**This Response to the Reviewers document provides a complete description of the changes that have been made in response to each individual reviewer comment. Reviewer comments are shown in plain text. Author responses are shown in blue text. All line numbers in the author responses refer to locations in the revised manuscript with changes marked.**

**Referee #1**

In their paper "Land Surface Model influence on the simulated climatologies of temperature and precipitation extremes in the WRF v.3.9 model over North America", the authors present an analysis of 4 WRF simulations run over North America over the same period, with each simulation using a different land surface model. Their study is clear and interesting, and serves not only to highlight the importance of land-atmosphere coupling on temperature and precipitation extremes, but also to demonstrate that while topography and the atmosphere itself do play a first order role in controlling extremes over land, the choice of land surface model itself can be a source of substantial spread.

The authors have done an excellent job addressing the concerns raised during the last round of revision, and the manuscript and figures have improved in their readability and clarity. I have only minor comments, noted below.

Line 290 / Figure 4: I feel the need to still push back at calling a region that covers the north-eastern corner of Canada and does not cover Greenland "Greenland". While I understand that the authors want to be consistent with the domains of a referenced study, they should at least acknowledge that the domain does not include Greenland and they're only calling it that for ease of comparison to Giorgi and Francisco (whose domain did actually include Greenland, unlike this study's domain).

**We have included a couple of lines clarifying this in the new version of the manuscript (See lines 287-289).**

Line 245: I appreciate the inclusion of more discussion on why each VAC category is occurring for each model (e.g. the low latent heat fluxes in CLM4 during winter). It would be even MORE helpful if the authors could identify what aspects of each LSM were resulting in the observed fluxes that control the VAC category, though they do allude to variations in representations of plants and snow.

**Differences between LSM components in the description of land cover affect the simulation of soil properties, such as albedo, evaporative resistance, and surface roughness. These soil properties play a key role in the computation of the energy and water fluxes at the land surface, and therefore in the simulation of near-surface conditions. We have included this explanation in lines 400-403.**

Methodological comment: I still think it would be really useful to see the difference in spread in results that occurs from using multiple ensemble members of a single LSM-WRF setup (e.g. multiple NOAH-WRF simulations) vs the spread across 4 single instantiations of 4 different LSM-WRF simulations. I

understand that that is computationally expensive, and don't think it is required for the publication of this study, but if the authors choose to further pursue the questions they raised in this study, explicitly quantifying the spread in an ensemble of single LSM-WRF runs and comparing that to the spread in WRF runs across multiple LSMs would (a) be interesting and (b) if the single LSM-WRF spread proves to be small, make the results presented in this study more robust.

**Thanks for the suggestion, it is indeed an interesting future line of investigation. However, considering the computational resources required for that ensemble, it maybe more efficient to perform those simulations for a shorter period of time (e.g. 5 or 10 years). We could compare that ensemble with the spread among the LSM configurations for that period of time and use the remaining computational resources to modify atmospheric parameterizations. Although we discussed the effect of atmospheric parameterizations on the simulations using references from the literature, its direct comparison with our simulations is also an interesting research line.**

Minor typos:

Line 85: "examinate" should be "examine"

**Done (line 85).**

Line 173/174: "increases" and "decreases" should be "increasing" and "decreasing"

**Corrected (lines 172).**

**Referee #2**

I believe the authors have sufficiently responded to my reviews. The changes they have made based on both reviewers have improved the paper, and beyond a few technical corrections the paper is ready for publication.

Line 25: confirm "that" all of these...

**Done (line 25).**

Line 34: of is typed two times, one needs to be removed.

**Thanks for catching that. It has been corrected (line 34).**

Line 49: I believe it should be snow cover, not snow covers.

**Agreed and changed (line 48).**

Paragraph staring on line 67: As ERA5 and MERRA2 have been published for a while now, a brief description of how their land surface components are treated should be included.

**We have included information about the ERA5 product and the MERRA2 product as requested (lines 71-75 and 81-82).**

Line 225: Would be better worded as: "agreeing in seasonality and broadly in the regional classification of energy and water limited areas (e.g. areas with high probability of episodes where atmospheric forcing or soil conditions control land-atmosphere interactions).

**Changed (line 222-224).**

I appreciate the inclusion of the discussion of how land-use differences and snow cover could lead to differences LA coupling in the models. I think, however, you need to be more careful with your language about cause/effect. e.g. on line 381 you say "representation of vegetation cover are causing". To really know if this was the cause i think sensitivity tests where the land cover type is changed in each model would need to be done. You could change the language to "likely play a role" or "influence" in these cases, but directly stating "cause" is beyond the scope of this analysis.

**We have gone through the manuscript addressing the reviewer concern (e.g. line 381, 398, and 556).**

Line 458: I am confused by the confusion of "ocean parameterizations" here. In these RCM simulations the ocean is just a boundary condition provided by the GCM. The way the boundary layer responds to the ocean and fluxes are calculated will be different, but those are not really ocean parameterizations. I suggest removing "ocean parameterizations" here and add it to "treatment of boundary conditions, including sea surface temperatures over the ocean"

**Agreed. We have modified these lines accordingly (see line 457).**

---

## Author Response (AR3)

**Response to Topical Editor Comments: Document for GMD-2020-86 by Almudena García-García, Francisco José Cuesta-Valero, Hugo Beltrami, Fidel González-Rouco, Elena García-Bustamante and Joel Finnis**

**Dear Dr. Garcia-Garcia -- thank you for addressing the reviewer comments. I believe the publications is ready for acceptance subject to some improvements to the code and data availability.**

**In particular, we would ask that you archive all relevant materials to the ZENODO archive associated with the project so as to maintain a persistent identifier associated with the code. I have checked with the executive editor and he indicates that even the WRF3.9 source code should be placed in the ZENODO archive.**

**We have updated the ZENODO archive to include the WRFV3.9 code, namelists, tables and the geofiles documents that are required to reproduce the simulations.**

**This prevents the link from failing in the case of a website reorganization. Similarly they have asked that the climdex code be placed within this archive as well, as cran.r-project.org is a software distribution system rather than a persistent archive.**

**The new version of the files in the ZENODO server contains the climdex package.**

**The NARR product is acceptable in its current form since the data is large enough that duplication is a problem. Finally, the CORDEX link is still to a search page. A reader cannot properly identify the correct CORDEX data from this link. In fact, CORDEX provides a recommended citation format here: https://prtal.enes.org/data/data-metadata-service/data-publication/copy_of_data-citation.**

**We have requested the suggested DOI following the instructions detailed in the provided webpage. However, they can't provide a DOI for the data we are using, because the WDCC archive does not include simulations of the NA-CORDEX. So we keep the reference to Mearns, L.O., et al., 2017 as suggested in the NA-CORDEX webpage (https://na-cordex.org/ ) and the website where we downloaded the data ( https://www.earthsystemgrid.org/search/cordexsearch.html ).**